# Kinetochore inactivation by expression of a repressive mRNA

Jingxun Chen[1†], Amy Tresenrider[1†], Minghao Chia[2], David T McSwiggen[3], Gianpiero Spedale[2], Victoria Jorgensen[1], Hanna Liao[1], Folkert Jacobus van Werven[2]*, Elçin Ünal[1,4]*

[1]Department of Molecular and Cell Biology, University of California, Berkeley, Berkeley, United States; [2]The Francis Crick Institute, London, United Kingdom; [3]Department of Molecular and Cell Biology, Li Ka Shing Center, University of California, Berkeley, Berkeley, United States; [4]The Paul F. Glenn Center for Aging Research, University of California, Berkeley, Berkeley, United States

*For correspondence:
folkert.vanwerven@crick.ac.uk (FJW);
elcin@berkeley.edu (EçÜ)

[†]These authors contributed equally to this work

Competing interests: The authors declare that no competing interests exist.

**Abstract** Differentiation programs such as meiosis depend on extensive gene regulation to mediate cellular morphogenesis. Meiosis requires transient removal of the outer kinetochore, the complex that connects microtubules to chromosomes. How the meiotic gene expression program temporally restricts kinetochore function is unknown. We discovered that in budding yeast, kinetochore inactivation occurs by reducing the abundance of a limiting subunit, Ndc80. Furthermore, we uncovered an integrated mechanism that acts at the transcriptional and translational level to repress *NDC80* expression. Central to this mechanism is the developmentally controlled transcription of an alternate *NDC80* mRNA isoform, which itself cannot produce protein due to regulatory upstream ORFs in its extended 5' leader. Instead, transcription of this isoform represses the canonical *NDC80* mRNA expression in *cis*, thereby inhibiting Ndc80 protein synthesis. This model of gene regulation raises the intriguing notion that transcription of an mRNA, despite carrying a canonical coding sequence, can directly cause gene repression.

DOI: https://doi.org/10.7554/eLife.27417.001

## Introduction

Cellular differentiation programs depend on temporally controlled waves of gene activation and inactivation. These waves in turn drive the morphogenetic events that ultimately transform one cell type into another. Differentiation models ranging from *Bacillus subtilis* sporulation to mouse embryogenesis have elucidated how transcription factor handoffs temporally activate the expression of gene clusters (*Errington, 2003*; *Zernicka-Goetz et al., 2009*). In comparison, much less is understood about how gene repression is coordinated with the transcription factor-driven waves of gene expression and how this inactivation is mechanistically achieved.

One critical morphogenetic event that relies on inactivation is the loss of kinetochore function during meiotic prophase. The kinetochore is a protein complex that binds to centromeric DNA and serves as the attachment site for spindle microtubules to mediate chromosome segregation (*Musacchio and Desai, 2017*) (*Figure 1A*). In multiple systems, it has been shown that kinetochores do not bind to microtubules in meiotic prophase (*Asakawa et al., 2005*; *Kim et al., 2013*; *Meyer et al., 2015*; *Miller et al., 2012*; *Sun et al., 2011*). Furthermore, this temporal inactivation is achieved through removal of the outer kinetochore, the site where microtubule attachments occur (*Asakawa et al., 2005*; *Kim et al., 2013*; *Meyer et al., 2015*; *Miller et al., 2012*; *Sun et al., 2011*) (*Figure 1B*). In the presence of a spindle, cells that fail to disassemble the outer kinetochore undergo catastrophic missegregation of meiotic chromosomes, underlying the essential nature of kinetochore downregulation during meiotic prophase (*Miller et al., 2012*). Importantly, the

**eLife digest** DNA stores the genetic information needed to make proteins and other molecules inside cells. To make a protein, cells use a particular section of DNA as a template to make molecules of messenger RNA (or mRNA for short), which are then translated into the corresponding protein.

Inside yeast, humans and other eukaryotic organisms, DNA is organized into structures called chromosomes. When these cells divide to make sex cells, such as egg cells and sperm, they undergo a process known as meiosis to make four daughter cells with only half as many chromosomes as the parent cell. During meiosis the parent cell's chromosomes need to be separated twice in quick succession. Large assemblies of proteins known as kinetochores are essential for this process. At the beginning of meiosis the kinetochores are inactive, which prevents the chromosomes from being separated too soon. Later on, the kinetochores are activated to allow the chromosomes to be separated. In budding yeast cells, control of kinetochore activity is achieved by regulating the levels of a single protein within the kinetochore known as Ndc80. It was not clear, however, how this regulation occurs.

Chen, Tresenrider et al. show that two yeast mRNAs with opposing activities are the key to regulating when Ndc80 is produced during meiosis. These two mRNAs carry the same protein-coding message but one mRNA is longer and contains an extension that prevents it from being translated into protein. The sole role of the longer mRNA is to prevent the shorter mRNA from being produced. The longer mRNA is only made in early meiosis, which prevents new Ndc80 proteins from being made during this time and results in the kinetochores being inactivated. The chromosomes only separate when production of the shorter mRNA returns later in meiosis and higher levels of Ndc80 protein allow the kinetochores to reform.

These findings demonstrate that even if an mRNA molecule does carry protein-coding information, it may not actually act as a messenger to make that protein, but instead, it can serve a regulatory role by blocking that protein's production. The key components involved in regulating Ndc80 during meiosis are found in many organisms ranging from yeast to humans, suggesting that similar mechanisms could be used to control proteins involved in other processes in cells.

DOI: https://doi.org/10.7554/eLife.27417.002

kinetochore is reactivated when the outer kinetochore reassembles upon transition from prophase to the meiotic divisions. How the initial removal and subsequent reassembly of the outer kinetochore is coordinated with the meiotic gene expression program is unknown.

Budding yeast provides a powerful model to address how the dynamic regulation of kinetochore function is integrated into the meiotic gene expression program. Entry into meiosis marks a clear cell-fate transition defined by the induction of Ime1, a master transcription factor. Ime1 activates the expression of genes involved in DNA replication and meiotic recombination (*Kassir et al., 1988*; *van Werven and Amon, 2011*). Successful completion of recombination, in turn, induces a second transcription factor Ndt80, which activates the expression of genes involved in meiotic divisions and gamete development (*Chu and Herskowitz, 1998*; *Xu et al., 1995*). Thus, the landmark morphogenetic events in budding yeast meiosis are coordinated by the relay between these two transcription factors. Furthermore, a high-resolution map of the gene expression waves that drive meiosis has been generated for budding yeast (*Brar et al., 2012*). Importantly, analysis of this dataset revealed that, of the 38 genes that encode kinetochore subunits, *NDC80* displays the most regulated expression pattern between meiotic prophase and the subsequent division phases (*Miller et al., 2012*).

Ndc80 is the namesake member of an evolutionarily conserved complex that forms the microtubule-binding interface of the outer kinetochore (*Tooley and Stukenberg, 2011*) (*Figure 1A*). Numerous lines of evidence indicate that the tight regulation of *NDC80* is essential for the timely function of kinetochores during meiosis. First, the decline of Ndc80 protein in meiotic prophase correlates with the dissociation of the outer kinetochore from the chromosomes (*Kim et al., 2013*; *Meyer et al., 2015*; *Miller et al., 2012*). Second, even though the other outer kinetochore subunits are expressed in meiotic prophase, they do not localize to the kinetochores (*Meyer et al., 2015*). Third, the subsequent increase in Ndc80 protein coincides with outer kinetochore reassembly

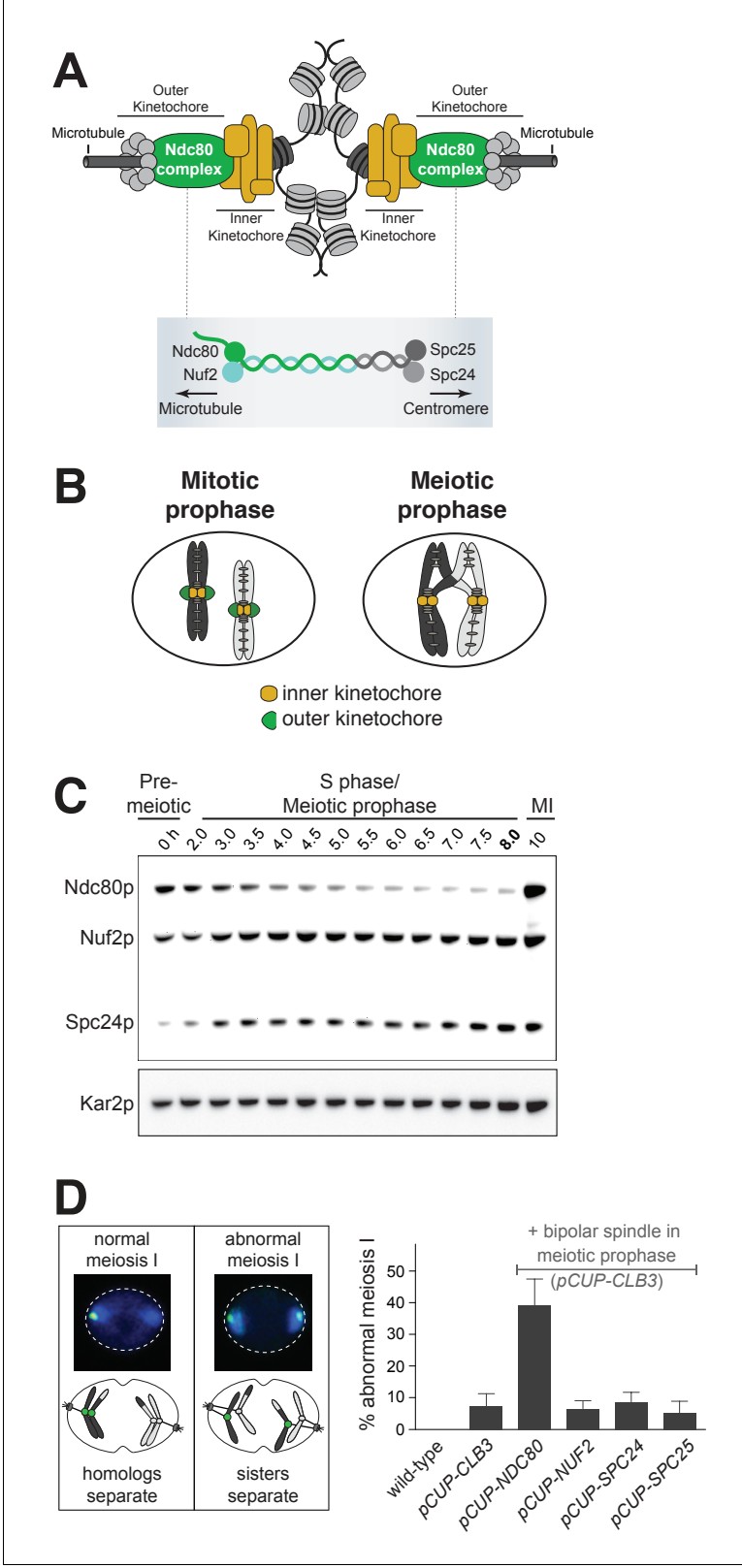

**Figure 1.** Kinetochore function is repressed during meiotic prophase due to limiting levels of Ndc80. (A–B) Schematics of kinetochore structure and dynamic behavior. (A) Top: kinetochores assembled on the centromere and attached to microtubules. Bottom: the Ndc80 complex. (B) During mitosis, the outer kinetochores are fully assembled, while in meiotic prophase, the outer kinetochores disassemble. (C) Ndc80, Nuf2, and Spc24 protein

*Figure 1 continued on next page*

*Figure 1 continued*

abundance in meiosis. Anti-V5 immunoblotting was performed at the indicated time points for three epitope-tagged subunits of the Ndc80 complex (Ndc80-3V5, Nuf2-3V5, and Spc24-3V5) in a single strain (UB4361). Using the *pGAL-NDT80 GAL4-ER* synchronization method (*Carlile and Amon, 2008*), cells were arrested in pachytene and then released 8 hr after the cells were transferred to SPO to allow progression into the meiotic divisions. One of the two repeated experiments is shown. (D) Sister chromatid segregation in wild type (UB4432), *pCUP-CLB3* (UB4434), *pCUP-CLB3 pCUP-NDC80* (UB880), *pCUP-CLB3 pCUP-NUF2* (UB4436), *pCUP-CLB3 pCUP-SPC24* (UB980), and *pCUP-CLB3 pCUP-SPC25* (UB885). A pair of sister chromatids of chromosome V was labeled with the centromeric TetO/TetR-GFP system (CENV-GFP). Left: A schematic depicting CENV-GFP dot localization in normal and abnormal meiosis I. In normal meiosis I, when homologous chromosomes segregate, a single GFP dot is present in one of the two nuclear masses of a binucleated cell. In abnormal meiosis I, when sister chromatids segregate, both nuclear masses of a binucleated cell contain a GFP dot. Right: The average fraction of binucleates that displayed sister chromatid segregation in meiosis I. Expression of Clb3 and each Ndc80 complex subunit (both regulated by the *pCUP* promoter) were co-induced by addition of $CuSO_4$ 6 hr after the cells were transferred to SPO. Concomitantly, cells were released from pachytene arrest by addition of β-estradiol. Cells were fixed 1 hr and 45 min after the release. The error bars represent the standard error of the mean from three independent experiments. 100 cells were counted per strain, per experiment.

DOI: https://doi.org/10.7554/eLife.27417.003

The following figure supplements are available for figure 1:

**Figure supplement 1.** Spc25 protein is present throughout meiotic prophase.

DOI: https://doi.org/10.7554/eLife.27417.004

**Figure supplement 2.** Over-expression of Ndc80, Nuf2, Spc24, and Spc25 during pro-metaphase I in *pCUP-NDC80-3V5* (UB880), *pCUP-NUF2-3V5* (UB12662), *pCUP-SPC24-3V5* (UB12543) and *pCUP-SPC25-3V5* (UB12547) strains, respectively.

DOI: https://doi.org/10.7554/eLife.27417.005

(*Meyer et al., 2015*; *Miller et al., 2012*). Finally, in the presence of a spindle, prophase misexpression of *NDC80* disrupts proper meiotic chromosome segregation (*Miller et al., 2012*). Together, these results indicate that *NDC80* regulation is necessary for the proper timing of kinetochore function in meiosis and highlight the importance of controlling Ndc80 protein levels during meiotic differentiation.

Here we uncovered how the timely function of kinetochores is achieved through the regulation of Ndc80 protein synthesis during budding yeast meiosis. This mechanism is based on the use of two *NDC80* mRNA isoforms, which have opposite functions and display distinct patterns of expression. In addition to the canonical protein-translating *NDC80* mRNA, we found that meiotic cells also expressed a 5'-extended *NDC80* isoform. Despite carrying the entire *NDC80* open reading frame (ORF), this alternate isoform cannot produce Ndc80 protein due to the presence of regulatory upstream ORFs (uORFs) in its extended 5' leader. Rather, its transcription plays a repressive role to inhibit transcription of the canonical *NDC80* mRNA in *cis* and thereby inhibit Ndc80 protein synthesis. Furthermore, we found that the expression of the 5'-extended isoform was activated by the meiotic initiator transcription factor Ime1. Upon exit from prophase, the mid-meiotic transcription factor Ndt80 activated the expression of the canonical *NDC80* mRNA isoform. Taken together, this study uncovers how *NDC80* gene repression is achieved and how inactivation and subsequent reactivation of the kinetochore is coordinated with the transcription factor-driven waves of meiotic gene expression.

## Results

### Ndc80 is the limiting component for kinetochore function in meiotic prophase

The Ndc80 complex consists of four subunits, namely Ndc80, Nuf2, Spc24, and Spc25 (*Figure 1A*). All the subunits other than Ndc80 persist in meiotic prophase (*Meyer et al., 2015*). Consistent with this report, we found that even in an extended meiotic prophase arrest, Ndc80 was the only subunit of its complex whose abundance decreased at this meiotic stage (*Figure 1C* and *Figure 1—figure*

*supplement 1*). Nuf2, Spc24, and Spc25 were all expressed, though it has been reported that these proteins fail to localize to the kinetochores during meiotic prophase (*Meyer et al., 2015*).

These observations raised the possibility that Ndc80 could be the limiting kinetochore subunit in meiosis. If correct, then the elevation of Ndc80 protein levels, but not the other subunits, should reactivate kinetochore function in meiotic prophase. To test this prediction, we overexpressed each of the Ndc80 complex subunits (*Figure 1—figure supplement 2*), in conjunction with the B-type cyclin Clb3, under an inducible *CUP1* promoter (*pCUP*). *CLB3* misexpression causes bipolar spindle assembly in meiotic prophase (*Miller et al., 2012*). In *pCUP-CLB3* cells, if kinetochores are functional in meiotic prophase, they attach to the spindle microtubules prematurely. These premature attachments, in turn, cause sister chromatid segregation in meiosis I, essentially disrupting proper meiotic chromosome segregation (*Miller et al., 2012*). When *NDC80* was overexpressed in *pCUP-CLB3* cells during meiotic prophase, over 30% of the cells displayed an abnormal segregation pattern in meiosis I. In contrast, misexpression of *CLB3* alone resulted in only a 7% segregation defect. Importantly, this defect was not further enhanced by the overexpression of *NUF2*, *SPC24* or *SPC25* (*Figure 1D*). Based on this observation, we conclude that kinetochore function is repressed in meiotic prophase due to limiting levels of Ndc80. Following prophase, Ndc80 becomes highly abundant during the meiotic divisions (*Miller et al., 2012*) (*Figures 1C*, 10 h time point), consistent with its role in facilitating chromosome segregation (*Wigge and Kilmartin, 2001*). Together, these results demonstrate that Ndc80 is the sole subunit of its complex that is tightly regulated during meiotic differentiation and strongly support the notion that *NDC80* downregulation and re-synthesis govern kinetochore functionality in meiosis.

## Two distinct *NDC80* transcript isoforms exist in meiosis

To dissect the molecular mechanism for the strict temporal regulation of the *NDC80* gene in meiosis, we first took advantage of the high-resolution RNA-seq and ribosome profiling dataset generated for budding yeast meiosis (*Brar et al., 2012*). Analysis of this dataset revealed the presence of meiosis-specific RNA-seq reads that extend to ~500 base pairs (bp) upstream of the *NDC80* ORF (*Figure 2A*). These reads appeared after meiotic entry and persisted until the end of meiosis, but were absent during vegetative growth (*Figure 2—figure supplement 1*, vegetative) or starvation (*Figure 2—figure supplement 1*, *MATa/MATa*).

To monitor the different RNA molecules generated from the *NDC80* locus, we performed northern blotting. In the absence of meiotic progression, when cells were subject to nutrient poor conditions, we detected only a single *NDC80* transcript throughout the starvation regime (no $CuSO_4$, *Figure 2—figure supplement 2A*). However, in cells undergoing synchronous meiosis, two distinct *NDC80* transcript isoforms became evident: a longer, meiosis-specific isoform, and a shorter isoform that was also present under non-meiotic conditions (*Figure 2B* and *Figure 2—figure supplement 2*). The longer isoform appeared after meiotic entry, persisted throughout meiotic prophase and gradually disappeared during the meiotic divisions. The shorter isoform was present in vegetative cells prior to meiotic entry, but was weakly expressed during S phase and meiotic prophase. Its abundance dramatically increased during the meiotic divisions (*Figure 2B* and *Figure 2—figure supplements 2B* and *3*). Interestingly, the Ndc80 protein levels were noticeably higher during the meiotic stages when the shorter transcript was the predominant isoform, but lower when the longer transcript was predominant (*Figure 2B*).

In addition to northern blotting, we used single molecule RNA fluorescence *in situ* hybridization (smFISH) to assess the cell-to-cell variability in transcript expression and subcellular localization of these two *NDC80* transcript isoforms. With two sets of probes that bind to the same region of *NDC80* ORF (odd/even probes), we verified that our smFISH could uniquely pair the FISH spots from these two probe sets with an accuracy of 88% (*Figure 2—figure supplement 4*), a value similar to what was reported previously (*Raj et al., 2008*). Furthermore, we confirmed that the number of cells analyzed per sample per experimental repeat (>95 cells) exceeded the minimal number of cells required to achieve a stable sampling average (*Figure 2—figure supplement 5*), and thus our sample size is large enough to reflect the population mean.

To differentiate between the two *NDC80* isoforms, we used another two sets of probes: one set (Q 670), conjugated to Quasar 670, is complementary to the sequences common between the short and the long isoforms. The other set (CF 590), conjugated to CAL Fluor Red 590, is unique to the long isoform. The long isoforms were identified as the spots where the signal from both probe sets

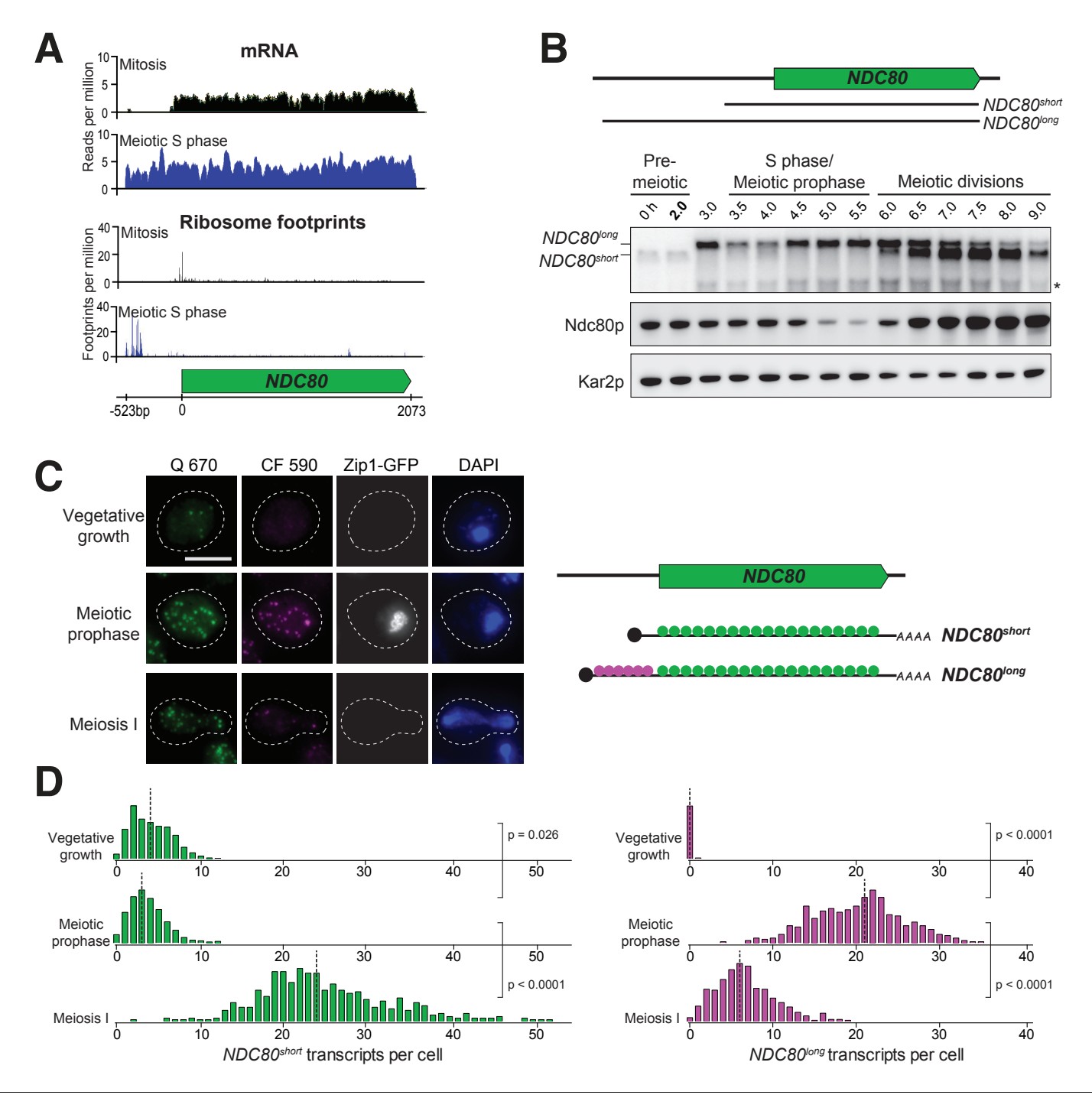

**Figure 2.** Two distinct *NDC80* transcripts are expressed during meiosis. (A) Ribosome profiling and mRNA-seq reads over the *NDC80* locus during vegetative growth (top track) or meiotic S phase (bottom track). Data are derived from (***Brar et al., 2012***). (B) *NDC80* mRNA isoforms and Ndc80 levels in meiosis. *NDC80^long^* and *NDC80^short^* levels were determined by northern blot, and Ndc80 level was determined by anti-V5 immunoblot at the indicated time points. To induce meiotic entry, *IME1* and *IME4* expression was induced in the strain UB1337 by addition of CuSO₄ 2 hr after cells were transferred to SPO. *SCR1*, loading control for northern blot. Kar2, loading control for immunoblot. One of the two repeated experiments is shown. * indicates a smaller RNA product, which likely represents a truncated form of *NDC80^long^*. (C) Representative smFISH images for *NDC80^long^* and *NDC80^short^* during vegetative growth and meiosis. Vegetative samples were taken when cells (UB8144) were growing exponentially in nutrient rich medium. Meiotic prophase samples were taken 6 hr after cells (UB8144) were transferred to SPO, a time when these cells were arrested in pachytene using the *pGAL-NDT80 GAL4-ER* system. Cells were then released by addition of β-estradiol, and meiosis I samples were taken 1.5 hr later. The Q 670 probes (shown in green) hybridize to the common region shared between *NDC80^long^* and *NDC80^short^*, whereas the CF 590 probes (shown in magenta)

*Figure 2 continued on next page*

Figure 2 continued

hybridize to the unique 5′ region of *NDC80^long^* (schematic is shown in the right panel). DNA was stained with DAPI (blue). Each cell was staged by its Zip1-GFP signal. Vegetative growth: Zip1-GFP negative. Meiotic prophase: Zip1-GFP positive. Meiosis I: Zip1-GFP negative and post *NDT80* induction. Images here and throughout are shown as the maximum-intensity projections of z-stacks. Scale bar: 5 µm. (**D**) Quantification of smFISH data shown in (**C**), graphed as the relative frequency histograms of cells with a given number of *NDC80^long^* and *NDC80^short^* transcripts per cell, using data pooled from three independent experiments. The dashed line indicates the median number of *NDC80^long^* and *NDC80^short^* transcripts per cell. Each histogram here and throughout was normalized so that the maximum bin height is the same across all histograms. A total number of 637 cells were analyzed for vegetative growth, 437 for meiotic prophase, and 491 for meiosis I. Two-tailed Wilcoxon Rank Sum test was performed between each pair of conditions as indicated by the bracket. Refer to **Supplementary file 1F** for a summary of the median transcript levels for all the smFISH experiments.
DOI: https://doi.org/10.7554/eLife.27417.006

The following figure supplements are available for figure 2:

**Figure supplement 1.** Ribosome profiling and mRNA-seq reads over the *NDC80* locus, during vegetative growth, starvation (*MATa/MATa*), and throughout meiosis.
DOI: https://doi.org/10.7554/eLife.27417.007

**Figure supplement 2.** During starvation, *NDC80^short^* and Ndc80 protein levels remain high, while *NDC80^long^* is not expressed.
DOI: https://doi.org/10.7554/eLife.27417.008

**Figure supplement 3.** Progression of cells through meiosis as determined by spindle morphology and DAPI staining.
DOI: https://doi.org/10.7554/eLife.27417.009

**Figure supplement 4.** Percentage of the colocalized or non-colocalized smFISH spots obtained using the odd and even smFISH probe sets.
DOI: https://doi.org/10.7554/eLife.27417.010

**Figure supplement 5.** Bootstrapping analysis performed for the data obtained from the odd and even probe sets.
DOI: https://doi.org/10.7554/eLife.27417.011

**Figure supplement 6.** smFISH quantification for *NDC80^long^* and *NDC80^short^* in pre-meiotic starvation and meiotic prophase.
DOI: https://doi.org/10.7554/eLife.27417.012

colocalized, whereas the short isoforms were identified as the spots with signal only from Q 670 (**Figure 2C and D**).

The smFISH analysis revealed that the expression of the two *NDC80* isoforms was temporally regulated. Vegetative cells expressed only the short *NDC80* isoform; fewer than 2% of these cells expressed the long isoform (**Figure 2C and D**). In meiotic prophase, a stage defined by the presence of the synaptonemal complex component Zip1, 100% of cells expressed the long isoform, and over 50% of them had more than 20 transcripts per cell. During the same stage, the level of the short isoform significantly decreased in comparison to its levels in vegetative growth (p=0.0260, two-tailed Wilcoxon Rank Sum test, **Figure 2D**) and pre-meiotic starvation (p=0.0090, **Figure 2—figure supplement 6**). As cells entered meiosis I, the level of the short isoform dramatically increased while that of the long isoform declined, in comparison to the levels of these isoforms during meiotic prophase (p<0.0001 for both *NDC80^short^* and *NDC80^long^* mRNAs, **Figure 2D**). Thus, the two *NDC80* isoforms have expression signatures specific to different cellular states.

In addition, the two *NDC80* isoforms localized to both the nucleus and cytoplasm (**Figure 2C**). We saw no evidence that the *NDC80^long^* isoform was solely retained in the nucleus; all of the Zip1-positive cells had at least one *NDC80^long^* mRNA localized outside of the DAPI-stained region. This localization pattern was consistent with the possibility that both transcripts were translated, as shown by ribosome profiling (**Figure 2A**, bottom panel) (**Brar et al., 2012**).

Altogether, the combined analyses of northern and western blotting, as well as smFISH, reveal two interesting trends: (1) In meiosis, the expression of the long and short *NDC80* isoforms are anti-correlated. (2) Ndc80 protein levels positively correlate with the presence of the short isoform and negatively correlate with the long isoform (**Figure 2B**).

## The long *NDC80* isoform is unable to produce Ndc80 protein due to translation of its upstream ORFs

The negative correlation between the longer *NDC80* isoform and Ndc80 protein levels suggested that this longer isoform was unable to support the synthesis of Ndc80 protein. In addition to the *NDC80* ORF, the longer isoform contains nine uORFs, each with an AUG start codon. The first six of these uORFs, those closest to the 5′ end of the mRNA, have ribosome profiling signatures consistent with them being translated in meiosis (**Figure 3—figure supplement 1**). Upstream start codons in

transcript leaders can capture scanning ribosomes to alternate reading frames, thereby restricting ribosome access to the main ORF (*Arribere and Gilbert, 2013*; *Calvo et al., 2009*; *Johnstone et al., 2016*).

We mutated the start codon of the first six uORFs (Δ6AUG) to test whether translation of the uORFs within the longer *NDC80* isoform represses translation of Ndc80 protein from this mRNA. In the Δ6AUG strain, the negative correlation between the long isoform and Ndc80 protein level persisted (*Figure 3*), potentially because translation of the remaining three uORFs could still repress translation of the ORF. Indeed, when all nine AUGs were mutated, Ndc80 protein became highly abundant during meiotic prophase, even though the long isoform remained the predominant *NDC80* transcript in these cells (*Figure 3*). These results demonstrate that although the longer isoform of *NDC80* contains the entire ORF, the presence of the uORFs in its 5′ leader prevents Ndc80 translation from this mRNA.

Next, we tested whether the repressive role of the uORFs resulted from the act of translation or the peptides encoded by these uORFs. We modified the long isoform, such that it still contained all the upstream AUG start codons, but each start codon was followed by a single amino acid and then immediately by a stop codon (*mini uORF*). Thus, this construct retained the translation ability of the uORFs but rendered them incapable of producing a peptide chain. We found that Ndc80 levels were still reduced during meiotic prophase in the *mini uORF* strain (*Figure 3*). Therefore, translation of the uORFs represses translation of the *NDC80* ORF from the long *NDC80* isoform, rendering this isoform unable to synthesize Ndc80 protein.

Our analyses so far demonstrate that the two *NDC80* mRNA isoforms differ with regards to their size and ORF coding capacity. The shorter isoform is capable of translating *NDC80* ORF. In contrast, although the longer isoform contains the entire ORF, it does not support Ndc80 synthesis. The coding information is not decoded from this isoform because uORF translation prevents ribosomes from accessing the actual ORF. To signify the unique features of each *NDC80* transcript isoform, we named the short mRNA *NDC80^{ORF}*, and the longer mRNA *NDC80^{luti}* for long un-decoded transcript isoform.

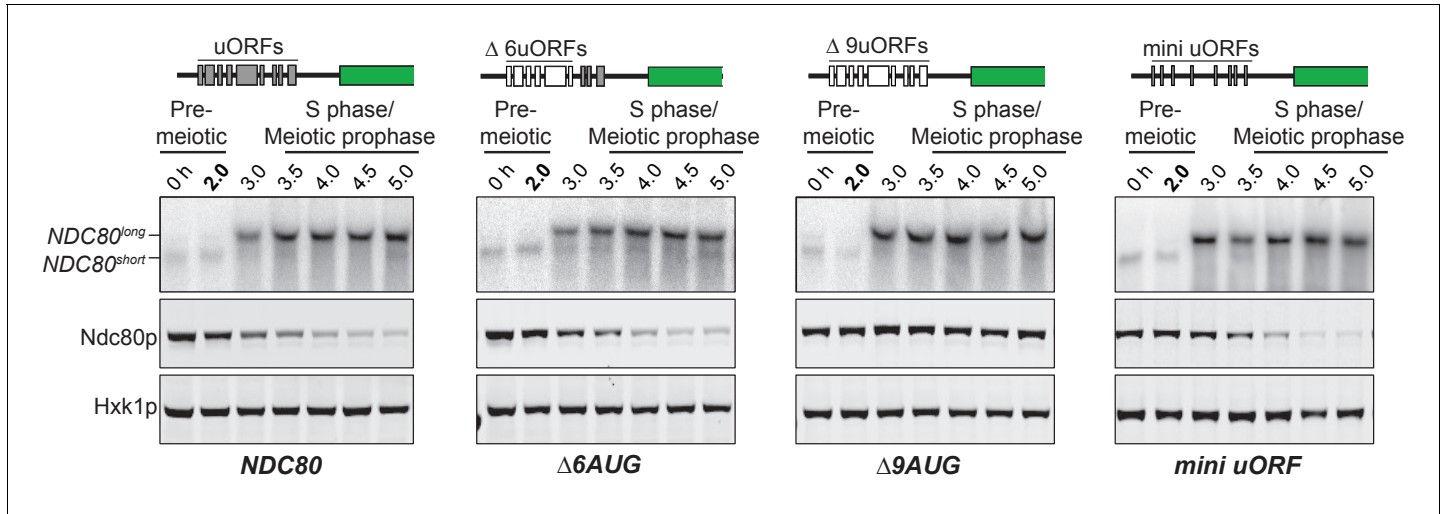

**Figure 3.** The longer *NDC80* mRNA isoform is unable to synthesize Ndc80 protein due to translation of its AUG uORFs. (**A**) *NDC80^{short}*, *NDC80^{long}*, and Ndc80 abundance during synchronous meiosis (as described in *Figure 2B*) in wild type (UB6190), Δ6AUG (UB6181), Δ9AUG (UB6183), and *mini uORF* (UB9243) strains. In the Δ6AUG and Δ9AUG strains, the first 6 or 9 uORF AUGs in the 5′ leader of *NDC80^{long}* were converted to AUCs, respectively. The *mini uORF* construct contained all 9 uORF start sites in the *NDC80^{long}* leader; however, the third codon of each of the 9 uORFs was mutated to a stop codon. One of the two repeated experiments is shown.

DOI: https://doi.org/10.7554/eLife.27417.013

The following figure supplement is available for figure 3:

**Figure supplement 1.** The first 6 AUG uORFs in the *NDC80^{long}* mRNA have ribosome footprint signatures consistent with them being translated.
DOI: https://doi.org/10.7554/eLife.27417.014

## $NDC80^{luti}$ expression in *cis* is necessary and sufficient to downregulate $NDC80^{ORF}$

Given that $NDC80^{luti}$ does not appear to produce Ndc80 protein, we set out to understand why meiotic cells express this mRNA isoform. Based on the observation that the expression levels of these two isoforms are anti-correlated, we posited that the transcription of $NDC80^{luti}$ represses $NDC80^{ORF}$. To test this hypothesis, we first eliminated $NDC80^{luti}$ production by deleting its promoter along with different portions of the $NDC80^{luti}$ transcript ($\Delta NDC80^{luti}$, **Figure 4—figure supplement 1**). As shown by northern blotting, $NDC80^{ORF}$ was detected during meiotic prophase in two different $\Delta NDC80^{luti}$ mutant strains (**Figure 4A** and **Figure 4—figure supplement 2**). Analysis of smFISH also confirmed that the level of $NDC80^{ORF}$ in $\Delta NDC80^{luti}$ cells significantly increased during meiotic prophase (**Figure 4B and C**, p=0.0004), with a median exceeding that of pre-meiotic cells (**Figure 2—figure supplement 6**). Accordingly, Ndc80 protein levels increased throughout meiotic prophase (**Figure 4A**).

Additionally, we inserted a termination sequence ~220 bp downstream of the $NDC80^{luti}$ transcription start site ($NDC80^{luti-Ter}$). We observed that, upon early termination of $NDC80^{luti}$, $NDC80^{ORF}$ mRNA and Ndc80 protein persisted in meiotic prophase (**Figure 4—figure supplement 3**). This observation suggests that continuous transcription through the $NDC80^{ORF}$ promoter is necessary for $NDC80^{ORF}$ repression. It also indicates that the repression of $NDC80^{ORF}$ is not due to competition between the $NDC80^{ORF}$ promoter and the $NDC80^{luti}$ promoter for RNA polymerase and the general transcription machinery. Altogether, we conclude that expression of the $NDC80^{luti}$ mRNA is required to repress the $NDC80^{ORF}$ transcript and reduce Ndc80 protein levels during meiotic prophase.

By what mechanism does $NDC80^{luti}$ reduce the steady-state level of $NDC80^{ORF}$? We posited that $NDC80^{luti}$ acts in *cis* based on other instances of overlapping transcription in budding yeast (**Bird et al., 2006**; **Martens et al., 2004**; **van Werven and Amon, 2011**). To test this, we engineered strains to have one wild type $NDC80^{luti}$ allele and another allele in which the promoter of $NDC80^{luti}$ has been deleted ($\Delta NDC80^{luti}$). In order to monitor Ndc80 protein levels, we inserted a 3V5 epitope as a C-terminal fusion to $NDC80$ in either the wild type or the $\Delta NDC80^{luti}$ allele. If $NDC80^{luti}$ functions in trans, then Ndc80-3V5 should be downregulated to the same extent in both strains. Instead, we found that Ndc80-3V5 was downregulated only when $NDC80^{luti}$ was generated on the same chromosome, directly upstream of $NDC80$-3V5 (**Figure 4D**, middle panel). This result demonstrates that $NDC80^{luti}$-mediated repression occurs in *cis*, since $NDC80^{luti}$ cannot reduce Ndc80 protein expression from a copy of $NDC80$ on another chromosome (**Figure 4D**, right panel). In the accompanying manuscript, Chia et al. revealed that this *cis*-acting mechanism is a result of alterations to the chromatin landscape across the $NDC80^{ORF}$ promoter caused by $NDC80^{luti}$ transcription (**Chia et al., 2017**).

Since $NDC80^{luti}$ is necessary to repress $NDC80^{ORF}$ during meiosis, we next investigated whether the $NDC80^{luti}$ leader is sufficient to regulate other genes in meiosis. We replaced the promoter and 5' leader of $NUF2$, the gene encoding the binding partner of Ndc80, with the promoter and 5' leader region of $NDC80^{luti}$ ($NDC80^{luti}$-$NUF2$). In wild type cells, a single $NUF2$ mRNA species was expressed in meiotic prophase, a stage when $NUF2$ mRNA levels and Nuf2 protein levels were stable (**Figure 5A and B**). In contrast, $NDC80^{luti}$-$NUF2$ cells expressed a longer mRNA ($NUF2^{luti}$) in meiotic prophase (**Figure 5A**), and the abundance of $NUF2^{ORF}$ transcripts was reduced by ~60% compared to that in the pre-meiotic stage (**Figure 5—figure supplement 1**), a reduction level similar to that of the Nuf2 protein (**Figure 5B**). This result demonstrates that the promoter and 5' leader sequence of $NDC80^{luti}$ is sufficient to downregulate another protein in meiotic prophase.

As $NDC80^{luti}$ expression is naturally restricted to meiosis, we tested whether the expression of $NDC80^{luti}$ was sufficient to downregulate $NDC80^{ORF}$ outside of meiosis. We artificially expressed $NDC80^{luti}$ during mitosis, a time when $NDC80^{luti}$ is naturally absent. We engineered strains in which the sole copy of the $NDC80$ gene had a modified upstream region, such that the endogenous promoter of $NDC80^{luti}$ was replaced by the inducible $GAL1$-$10$ promoter ($pGAL$-$NDC80^{luti}$). This alteration had minimal effect on cell growth (Figure 8C, uninduced), suggesting that $NDC80^{ORF}$ transcript and Ndc80 protein expression is largely unaffected in the absence of induction. In wild type cells synchronously progressing through the mitotic cell cycle, a single mRNA isoform, $NDC80^{ORF}$, was present at all stages (**Figure 5C**, left panel). In contrast, the $NDC80^{ORF}$ transcript became undetectable in $pGAL$-$NDC80^{luti}$ cells one hour after $NDC80^{luti}$ induction (**Figure 5C**, right

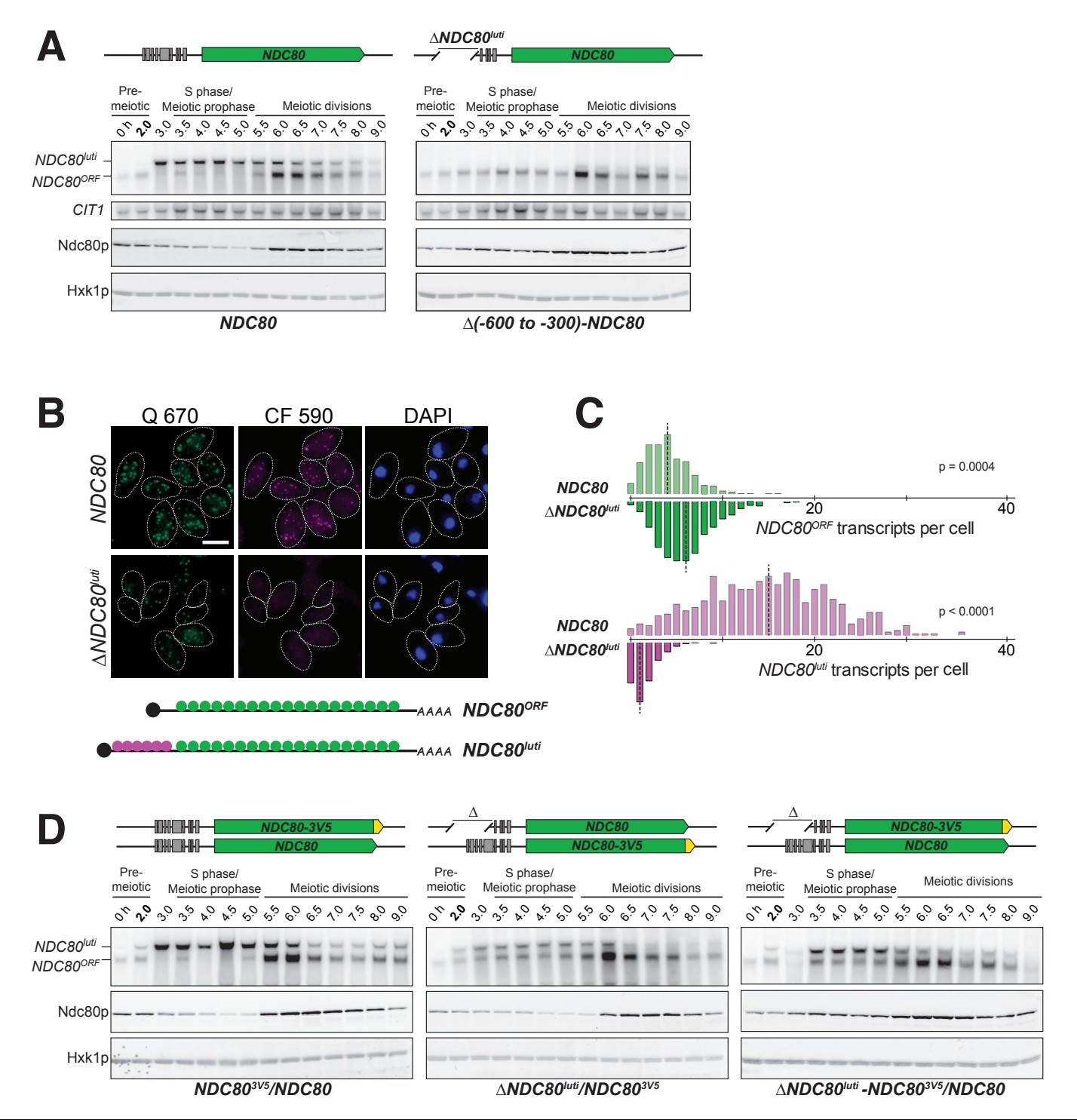

**Figure 4.** *NDC80*[*luti*] is necessary to downregulate *NDC80*[*ORF*]. (**A**) *NDC80*[*ORF*], *NDC80*[*luti*], and Ndc80 abundance during synchronous meiosis (as described in ***Figure 2B***) in wild type cells (FW1902) and in Δ*NDC80*[*luti*] cells (FW1871), in which 300–600 bp upstream of the Ndc80 translation start site were deleted. Ndc80 level was determined by anti-V5 immunoblot. *CIT1*, loading control for northern blot. Hxk1, loading control for immunoblot. One of the two repeated experiments is shown. (**B**) Representative smFISH images for *NDC80*[*luti*] and *NDC80*[*ORF*] during meiotic prophase in wild type cells (UB6190) and in Δ*NDC80*[*luti*] cells (UB6079), in which 479–600 bps upstream of the Ndc80 translation start site were deleted. This deletion construct was used, as opposed to the (−600 to −300) deletion, because this construct retains all the binding sites for the CF 590 probes (bind to the unique region of *NDC80*[*luti*]). Samples were taken 2 hr after *IME1* and *IME4* induction in a synchronous meiosis and hybridized with the Q 670 probes (bind to the common region of *NDC80*[*luti*] and *NDC80*[*ORF*], shown in green) and the CF 590 probes (shown in magenta), as in ***Figure 2C***. DNA was stained with DAPI
*Figure 4 continued on next page*

Figure 4 continued

(blue). Scale bar: 5 µm. (C) Quantification of smFISH data shown in (B), graphed as the relative frequency histograms of cells with a given number of $NDC80^{luti}$ and $NDC80^{ORF}$ transcripts per cell, using data pooled from three independent experiments. The dashed line indicates the median number of $NDC80^{luti}$ and $NDC80^{ORF}$ transcripts per cell. A total number of 611 cells were analyzed for wild type and 649 for $\Delta NDC80^{luti}$. Two-tailed Wilcoxon Rank Sum test was performed for $NDC80^{ORF}$ and $NDC80^{luti}$, respectively, comparing wild type with $\Delta NDC80^{luti}$ during meiotic prophase. (D) $NDC80^{luti}$ represses $NDC80^{ORF}$ expression in *cis*. Meiosis was induced and samples were collected and processed as in (A). Ndc80 level was determined by anti-V5 immunoblot. Hxk1, loading control. Three yeast strains were used in this experiment: 1) a strain (FW1900) with one $NDC80$-V5 allele and one wild type $NDC80$ allele (left), 2) a strain (FW1899) with one $NDC80$-3V5 allele and one $\Delta NDC80^{luti}$ allele, in which 300–600 bp upstream of the Ndc80 translation start site were deleted (middle), and 3) a strain (FW1923) with one $\Delta NDC80^{luti}$-$NDC80$-3V5 allele, which has the aforementioned 300–600 bps deletion, and one wild type $NDC80$ allele (right). One of the two repeated experiments is shown.

DOI: https://doi.org/10.7554/eLife.27417.015

The following figure supplements are available for figure 4:

**Figure supplement 1.** Annotated upstream intergenic region of the $NDC80$ locus and engineered mutations used in this study.

DOI: https://doi.org/10.7554/eLife.27417.016

**Figure supplement 2.** Expression of $NDC80^{luti}$ is necessary for $NDC80^{ORF}$ downregulation.

DOI: https://doi.org/10.7554/eLife.27417.017

**Figure supplement 3.** Premature termination of $NDC80^{luti}$ prevents $NDC80^{ORF}$ downregulation.

DOI: https://doi.org/10.7554/eLife.27417.018

panel). Four hours after induction, Ndc80 protein levels were reduced to 20% of the initial level, while in wild type cells it was increased to 116% (*Figure 5D*). Based on these data, we conclude that $NDC80^{luti}$ expression is sufficient to repress $NDC80^{ORF}$ outside of meiosis. The reduction in $NDC80^{ORF}$ expression, in turn, causes reduced synthesis of Ndc80 protein, thus essentially turning off the $NDC80$ gene.

## Master meiotic transcription factors Ime1 and Ndt80 regulate $NDC80^{luti}$ and $NDC80^{ORF}$ expression, respectively

Since the timely expression of $NDC80^{luti}$ and $NDC80^{ORF}$ is crucial to establish the temporal pattern of Ndc80 protein levels in meiosis, we next investigated which transcription factors directly control $NDC80^{luti}$ and $NDC80^{ORF}$ expression. In *S. cerevisiae*, meiotic gene expression is orchestrated by two master transcription factors: Ime1 and Ndt80 (*Chu and Herskowitz, 1998*; *Kassir et al., 1988*; *Xu et al., 1995*). Diploid *MATa/MATα* cells initiate meiosis by expressing *IME1* in response to nutrient deprivation (*van Werven and Amon, 2011*). Interestingly, *IME1* expression correlated with the time of $NDC80^{luti}$ expression, suggesting that Ime1 might regulate $NDC80^{luti}$ transcription. Indeed, deletion of *IME1* abolished $NDC80^{luti}$ production and resulted in persistent levels of $NDC80^{ORF}$ transcript and Ndc80 protein (*Figure 6A* and *Figure 6—figure supplement 1*).

Ime1 does not directly bind to DNA, but functions as a co-activator for Ume6 (*Washburn and Esposito, 2001*). In the absence of Ime1, Ume6 represses early meiotic genes in mitosis by binding to a consensus site called the upstream repressive sequence (URS1) in the promoters of these genes. Upon meiotic entry and subsequent interaction with Ime1, the Ume6-Ime1 complex activates the transcription of these early meiotic genes (*Bowdish et al., 1995*; *Park et al., 1992*). Given the close relationship between Ime1 and Ume6, we inspected the 5' intergenic region of $NDC80$ and identified a consensus site for Ume6 583 bp upstream of the Ndc80 translation start site (*Figure 6B* and *Figure 4—figure supplement 1*), within the $NDC80^{luti}$ promoter. ChIP analysis revealed that Ume6 binding was enriched over the predicted URS1 site in mitosis and early meiosis (*Figure 6C* and *Figure 6—figure supplement 2*), whereas Ume6 binding was undetectable within the $NDC80^{ORF}$ promoter (*Figure 6C* and *Figure 6—figure supplement 2*). Deletion of the URS1 site (*ndc80-urs1Δ*) completely abolished Ume6 binding to the $NDC80^{luti}$ promoter (*Figure 6C*), but did not affect another Ime1-Ume6 target gene *IME2* (*Figure 6—figure supplement 3*). Consistent with the role of Ume6 as a transcriptional repressor in mitosis, deletion of the URS1 site resulted in leaky expression of $NDC80^{luti}$ during vegetative growth (*Figure 6D and E*, p<0.0001) and reduced expression of $NDC80^{ORF}$ (*Figure 6E*, p=0.0057). Abolishing Ume6 binding eliminated strong induction of $NDC80^{luti}$ in meiosis (*Figure 6G and H*, p<0.0001), causing moderately increased levels of $NDC80^{ORF}$ transcript by northern blot and Ndc80 protein in meiotic prophase (*Figure 6I*). We did not detect significant increase in $NDC80^{ORF}$ in the *urs1Δ* cells by smFISH (*Figure 6H*), likely due to

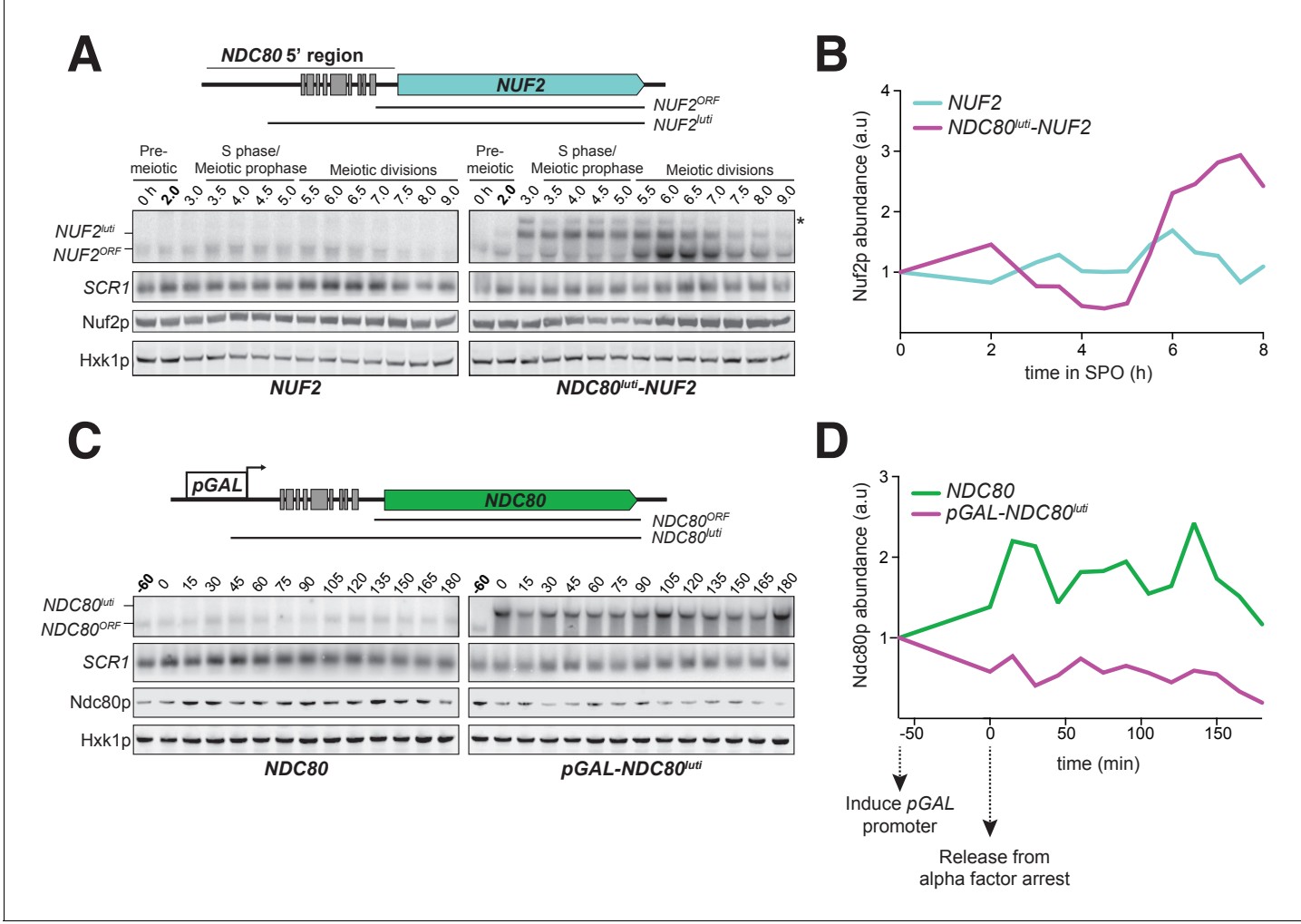

**Figure 5.** *NDC80<sup>luti</sup>* is sufficient to downregulate *NDC80<sup>ORF</sup>*. (**A**) A luti-mRNA is produced by the *NDC80<sup>luti</sup>-NUF2* fusion construct (*NUF2<sup>luti</sup>*) in meiosis. To generate the *NDC80<sup>luti</sup>-NUF2* construct, the promoter and leader sequence of *NDC80<sup>luti</sup>* (1000 bps directly upstream of the *NDC80* ORF start site) was placed immediately upstream of the *NUF2* coding region. *NUF2<sup>luti</sup>* and *NUF2<sup>ORF</sup>* expression was detected by northern blot, and Nuf2 was detected by anti-V5 immunoblot. *SCR1*, loading control for northern blot. Hxk1, loading control for immunoblot. Samples were taken when the wild type (UB5103) and *NDC80<sup>luti</sup>-NUF2* (UB5101) cells were undergoing synchronous meiosis. * indicates a band of unknown origin. One of the two repeated experiments is shown. (**B**) Quantification of Nuf2 protein abundance from the experiment shown in (**A**). For each time point, Nuf2 signal was first normalized to Hxk1. This normalized value was set to 1 for the 0 hr time point (t$_0$), and all the subsequent time points were calibrated relative to t$_0$. (**C**) *NDC80<sup>ORF</sup>*, *NDC80<sup>luti</sup>*, and Ndc80 levels when *NDC80<sup>luti</sup>* is expressed in synchronous mitosis. *MATa* wild type control (UB2389) and *pGAL-NDC80<sup>luti</sup>* (UB2388) cells, both harboring the Gal4-ER fusion protein, were arrested in G1 with α-factor. *pGAL* expression was induced 2 hr later by addition of β-estradiol (−60 min). One hour after the β-estradiol addition (0 min), cells were released from G1 arrest. One of the two repeated experiments is shown. (**D**) Quantification of Ndc80 abundance from the experiment shown in (**C**). For each time point, Ndc80 signal was first normalized to Hxk1. This normalized value was set to one for the first time point at −60 min (t$_{-60}$, the time of β-estradiol addition) and all the subsequent time points were then calibrated relative to t$_{-60}$.

DOI: https://doi.org/10.7554/eLife.27417.019

The following figure supplement is available for figure 5:

**Figure supplement 1.** Quantification of *NUF2* mRNA abundance from experiment shown in *Figure 5A*.

DOI: https://doi.org/10.7554/eLife.27417.020

technical reasons (See Materials and Methods). We conclude that similar to early meiotic genes, Ime1 and Ume6 directly regulate the transcription of *NDC80<sup>luti</sup>*.

The second key meiotic transcription factor, Ndt80, is required for meiotic chromosome segregation and spore formation (*Chu and Herskowitz, 1998*; *Xu et al., 1995*). Expression of *NDT80* occurs shortly before the reappearance of *NDC80<sup>ORF</sup>* transcript. Within the budding yeast lineage, an

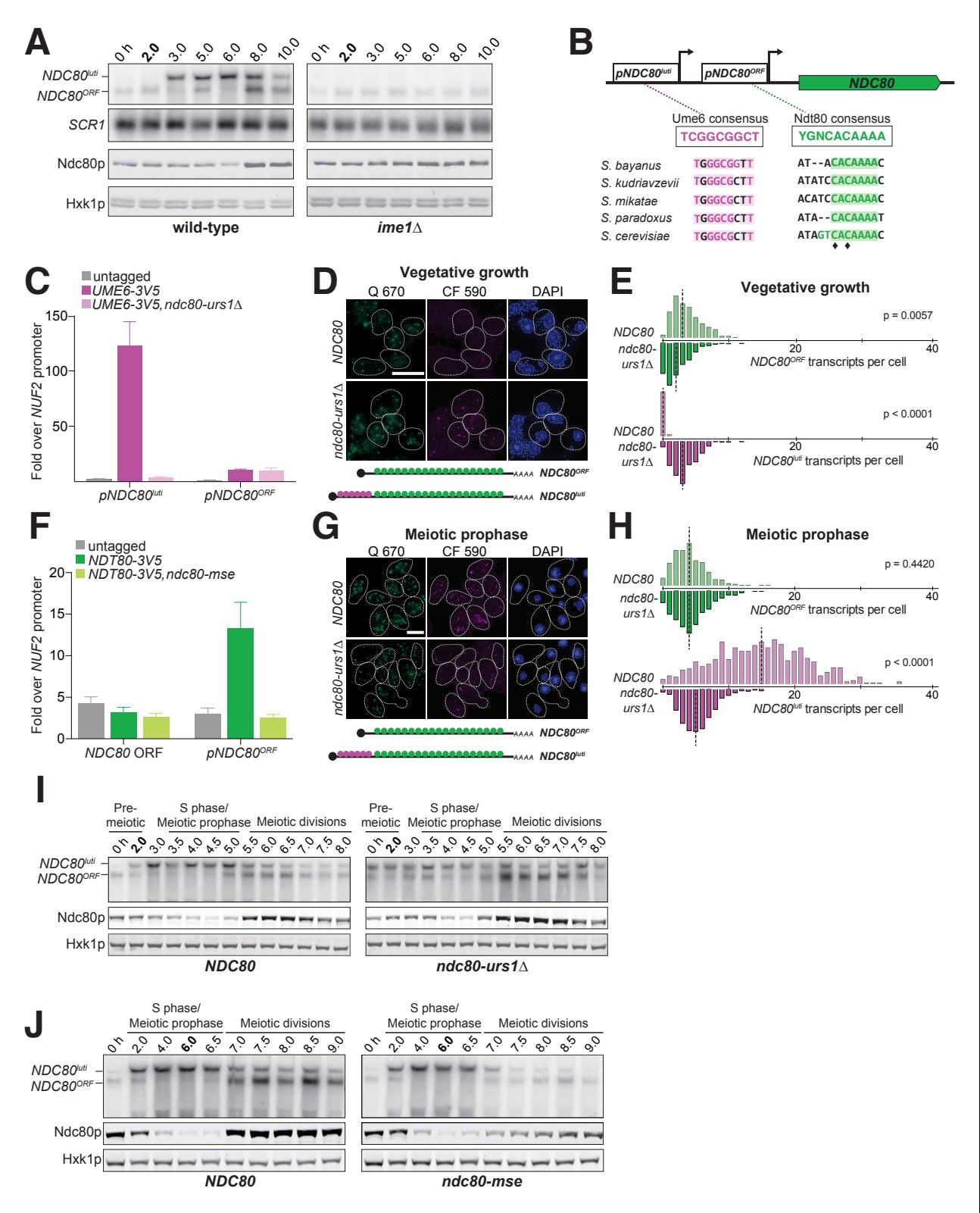

**Figure 6.** The meiosis-specific transcription factors Ime1 and Ndt80 regulate *NDC80luti* and *NDC80ORF*, respectively. (**A**) *NDC80ORF*, *NDC80luti*, and Ndc80 abundance during meiosis in *pCUP-IME1 pCUP-IME4* (FW1902) and *pCUP-IME4 ime1Δ* (FW3058) cells. Expression from the *pCUP* promoter was induced 2 hr after cells were transferred to SPO. One of the two repeated experiments is shown. (**B**) Putative Ume6 (URS1) and Ndt80 (MSE) binding sites are present in the intergenic region upstream of *NDC80*. Colored bases match the consensus binding sequences. Highlighted areas indicate the
*Figure 6 continued on next page*

*Figure 6 continued*

conserved regions across all five *Saccharomyces* species by Clustal analysis (RRID:SCR_001591). The black diamonds indicate the two sites mutated from C to A in the *ndc80-mse* strain. (C) Ume6-3V5 chromatin immunoprecipitation in untagged (UB2531), *UME6-3V5* (UB3301), and *UME6-3V5 ndc80-urs1Δ* (UB6760) strains. Cells were harvested after overnight growth in BYTA. The DNA fragments recovered from the Ume6-3V5 ChIP were quantified by qPCR using two primer pairs: one specific for the $NDC80^{luti}$ promoter and one specific for the $NDC80^{ORF}$ promoter. Enrichment at these loci was normalized to the signal from the *NUF2* promoter, to which Ume6 does not bind. The mean fold enrichment over the *NUF2* promoter from three independent experiments, as well as the standard error of the mean, is displayed. (D) Representative smFISH images for $NDC80^{luti}$ and $NDC80^{ORF}$ during vegetative growth in wild type (UB5875) and *ndc80-urs1Δ* (UB5473) strains. Cells were grown in nutrient rich medium to exponential phase. Samples were fixed and hybridized with the Q 670 probes (bind to the common region of $NDC80^{luti}$ and $NDC80^{ORF}$, shown in green) and the CF 590 probes (bind to the unique region of $NDC80^{luti}$, shown in magenta) as in *Figure 2C*. DNA was stained with DAPI (blue). Scale bar: 5 μm. (E) Quantification of (D), graphed as the relative frequency histograms of cells with a given number of $NDC80^{luti}$ and $NDC80^{ORF}$ transcripts per cell, using data pooled from three independent experiments. The dashed line indicates the median number of $NDC80^{luti}$ and $NDC80^{ORF}$ transcripts per cell. A total number of 490 cells were analyzed for wild type and 427 for *ndc80-urs1Δ*. Two-tailed Wilcoxon Rank Sum test was performed for $NDC80^{ORF}$ and $NDC80^{luti}$, respectively, comparing wild type with *ndc80-urs1Δ* in vegetative growth. (F) Ndt80-3V5 chromatin immunoprecipitation in untagged (UB7997), *NDT80-3V5* (UB7999), and *NDT80-3V5 ndc80-mse* strains (UB7496). After 5 hr in SPO, *NDT80* expression was induced with β-estradiol. One hour after Ndt80 induction, cells were fixed with formaldehyde and chromatin extracts were prepared. The recovered DNA fragments were quantified by qPCR using two primer pairs: one specific for the $NDC80^{ORF}$ promoter ($pNDC80^{ORF}$) and one specific to the *NDC80* coding region (*NDC80* ORF). Enrichment at these loci was normalized to the signal from the *NUF2* promoter, to which Ndt80 does not bind. The mean fold enrichment over the *NUF2* promoter from three independent experiments, as well as the standard error of the mean, is displayed. (G) Representative smFISH images for $NDC80^{luti}$ and $NDC80^{ORF}$ during meiotic prophase in wild type (UB6190) and *ndc80-urs1Δ* (UB6075) strains. Samples were taken 2 hr after *IME1* and *IME4* induction in a synchronous meiosis experiment and processed as in *Figure 2C*. Scale bar: 5 μm. Note: the image for wild type is the same as the one shown in *Figure 4B*. (H) Quantification of (G), graphed as relative frequency histograms as in (E). A total number of 611 cells were analyzed for wild type and 668 for *ndc80-urs1Δ*. Two-tailed Wilcoxon Rank Sum test was performed for $NDC80^{ORF}$ and $NDC80^{luti}$, respectively, comparing wild type with *ndc80-urs1Δ* during meiotic prophase. Note: the histograms for the wild type cells are the same as those shown in *Figure 4C*. (I) $NDC80^{ORF}$, $NDC80^{luti}$, and Ndc80 levels during synchronous meiosis (as described in *Figure 2B*) in wild type cells (UB6190) and *ndc80-urs1Δ* cells (UB6075). (J) $NDC80^{ORF}$, $NDC80^{luti}$, and Ndc80 level during meiosis in wild type (UB4074) and *ndc80-mse* (UB3392) strains. Both strains harbor the *pGAL-NDT80 GAL4-ER* system. Cells were transferred to SPO at 0 hr and released from pachytene arrest at 6 hr by addition of β-estradiol.

DOI: https://doi.org/10.7554/eLife.27417.021

The following figure supplements are available for figure 6:

**Figure supplement 1.** Quantification of the $NDC80^{ORF}$ transcript abundance shown in *Figure 6A*, a time course comparing the *pCUP-IME1 pCUP-IME4* strain (FW1902) during meiosis with the *pCUP-IME4 ime1Δ* strain (FW3058).
DOI: https://doi.org/10.7554/eLife.27417.022

**Figure supplement 2.** Ume6 is enriched at the $NDC80^{luti}$ promoter but not the *NDC80* coding region before and during early meiosis.
DOI: https://doi.org/10.7554/eLife.27417.023

**Figure supplement 3.** Deletion of the putative URS1 site upstream of *NDC80* does not affect Ume6 enrichment on the *IME2* promoter.
DOI: https://doi.org/10.7554/eLife.27417.024

**Figure supplement 4.** Deletion of the putative MSE site upstream of *NDC80* does not affect Ndt80 enrichment on the *MAM1* promoter.
DOI: https://doi.org/10.7554/eLife.27417.025

Ndt80 consensus site, called the mid-sporulation element (MSE), was identified at 184 bp upstream of the Ndc80 translation start site (*Figure 6B* and *Figure 4—figure supplement 1*), within the $NDC80^{ORF}$ promoter. One hour after Ndt80 expression was induced in the *pGAL-NDT80 GAL4-ER* system, Ndt80 binding was enriched over the predicted MSE by ChIP analysis; moreover, mutations in the MSE (*ndc80-mse*) led to a complete loss of Ndt80 enrichment (*Figure 6F*), but did not affect another Ndt80 target gene *MAM1* (*Figure 6—figure supplement 4*). Furthermore, the defect in Ndt80 binding to the $NDC80^{ORF}$ promoter reduced both $NDC80^{ORF}$ transcript and Ndc80 protein levels during the meiotic divisions (*Figure 6J*). These results demonstrate that Ndt80 directly induces $NDC80^{ORF}$ expression after meiotic prophase, and this timely induction of $NDC80^{ORF}$ elevates the levels of Ndc80 protein prior to the meiotic divisions.

## Temporal regulation of $NDC80^{luti}$ and $NDC80^{ORF}$ expression is essential for the proper timing of kinetochore function

Since Ndc80 appears to be the limiting subunit of the kinetochore, we posited that the regulated expression of $NDC80^{luti}$ and $NDC80^{ORF}$ serves to inactivate and reactivate kinetochores, respectively, through modulating Ndc80 protein levels. In budding yeast, kinetochores are inactive in meiotic prophase (*Miller et al., 2012*, and *Figure 1D*), but they can be activated upon Ndc80 overexpression (*Miller et al., 2012*, and *Figure 1D*). We asked whether functional kinetochores

could also be generated in meiotic prophase if cells failed to express $NDC80^{luti}$ ($\Delta NDC80^{luti}$) or expressed a version of $NDC80^{luti}$ that could translate Ndc80 protein ($\Delta 9AUG$). Both conditions caused an increase in Ndc80 levels in meiotic prophase (*Figures 3* and *4A*). Using the same assay described in *Figure 1D*, we observed that over 50% of the $\Delta NDC80^{luti}$ or $\Delta 9AUG$ cells displayed abnormal chromosome segregation in meiosis I (*Figure 7A*), suggesting premature kinetochore activity in meiotic prophase. The extent of this phenotype was indistinguishable from that when Ndc80 was overexpressed in meiotic prophase (*pCUP-NDC80*) (*Figure 7A*). Therefore, repression of $NDC80^{ORF}$ by $NDC80^{luti}$ transcription is crucial to inhibit untimely kinetochore function during meiotic prophase.

Functional kinetochores must be present after meiotic prophase to faithfully execute chromosome segregation during the two meiotic divisions. Since Ndc80 protein levels become nearly undetectable during prophase (*Figure 1C*), Ndc80 must be resynthesized to restore the ability of kinetochores to interact with microtubules upon exit from prophase. This resynthesis relies on the transcription factor Ndt80 to induce transcription of $NDC80^{ORF}$ (*Figure 6F and J*). To test the significance of Ndt80-dependent induction of $NDC80^{ORF}$ in meiosis, we monitored the segregation pattern of chromosome V in cells with a mutated Ndt80 binding site in the $NDC80^{ORF}$ promoter (*ndc80-mse*). Only 1% of wild type cells missegregated chromosome V, whereas 98% of the *ndc80-mse* cells failed to properly segregate this chromosome (*Figure 7B*), suggesting that kinetochores are not functional in *ndc80-mse* cells. In support of this conclusion, in *ndc80-mse* cells, elongated bipolar spindles (over 2 μm) appeared earlier and persisted longer than in wild type cells (*Figure 7C*), a phenomenon consistent with defective microtubule-kinetochore attachments (*Wigge et al., 1998*; *Wigge and Kilmartin, 2001*). Additionally, the abundance of short meiosis II spindles (less than 2 μm) was reduced in the *ndc80-mse* cells (*Figure 7D*), and at the end of meiosis, more than four nuclei were often observed (representative images shown in *Figure 7B*). The *ndc80-mse* mutation also severely affected the sporulation efficiency (*Figure 7—figure supplement 1*). All of these results demonstrate that Ndt80-dependent induction of $NDC80^{ORF}$ is essential for re-establishing kinetochore function to mediate meiotic chromosome segregation.

Unlike $NDC80^{ORF}$ transcript, $NDC80^{luti}$ is absent in vegetative growth due to repression by Ume6 (*Figure 6D and E*). We hypothesized that $NDC80^{luti}$ is repressed during the mitotic cell cycle because its expression could inactivate kinetochore function (*Figure 5C and D*). Indeed, when the Ume6 repressor-binding site within the $NDC80^{luti}$ promoter was deleted (*urs1Δ*), these cells grew similar to wild type cells at 30°C, but they had a severe growth defect at 37°C due to reduced Ndc80 levels (*Figure 8A and B*). Thus, the repression of $NDC80^{luti}$ by Ume6 is critical for the fitness of mitotically dividing cells.

When $NDC80^{luti}$ was strongly induced in vegetative growth using the inducible *GAL1-10* promoter, these cells had a severe growth defect (*Figure 8C*). This defect was rescued by a second copy of *NDC80* at an ectopic locus, consistent with the notion that $NDC80^{luti}$-mediated repression of $NDC80^{ORF}$ occurs in *cis* (*Figure 8C* and *Figure 4D*). Cell death was also rescued by silencing the *pGAL*-induced $NDC80^{luti}$ expression using CRISPRi (*Qi et al., 2013*) (*Figure 8D*), presumably due to the activation of the $NDC80^{ORF}$ promoter in the absence of $NDC80^{luti}$ transcription. Induction of the uORF-free $NDC80^{luti}$ ($\Delta 9AUG$) caused no appreciable growth defect (*Figure 8C*), consistent with the observation that the $\Delta 9AUG$ cells could express Ndc80 protein (*Figure 3*).

The inducible nature of the *GAL1-10* promoter allowed us to directly test whether the growth defect associated with the mitotic $NDC80^{luti}$ expression arose from defects in kinetochore function. We performed fluorescence microscopy to track spindle length (Spc42-mCherry) and chromosome segregation (CENV-GFP dots). Cells expressing $NDC80^{luti}$ displayed a range of kinetochore-microtubule attachment defects (*Figure 8E*, bottom panel). In cells with separated spindle pole bodies, ~30% of the cells expressing $NDC80^{luti}$ had metaphase spindles ($\leq 2$ μm) improperly localized to either the bud or the bud neck, whereas only 3% of the wild type cells displayed this phenotype (*Figure 8F*). Furthermore, in cells expressing $NDC80^{luti}$, an abnormal distribution of spindle length was observed, characteristic of a metaphase arrest (*Figure 8—figure supplement 1*). Spindle elongation was also observed prior to chromosome capture, suggesting improper kinetochore function (*Figure 8G*). Collectively, these analyses revealed that the strict temporal regulation of $NDC80^{luti}$ and $NDC80^{ORF}$ transcription in both mitosis and meiosis is essential to ensure the proper timing of kinetochore function and high fidelity chromosome segregation.

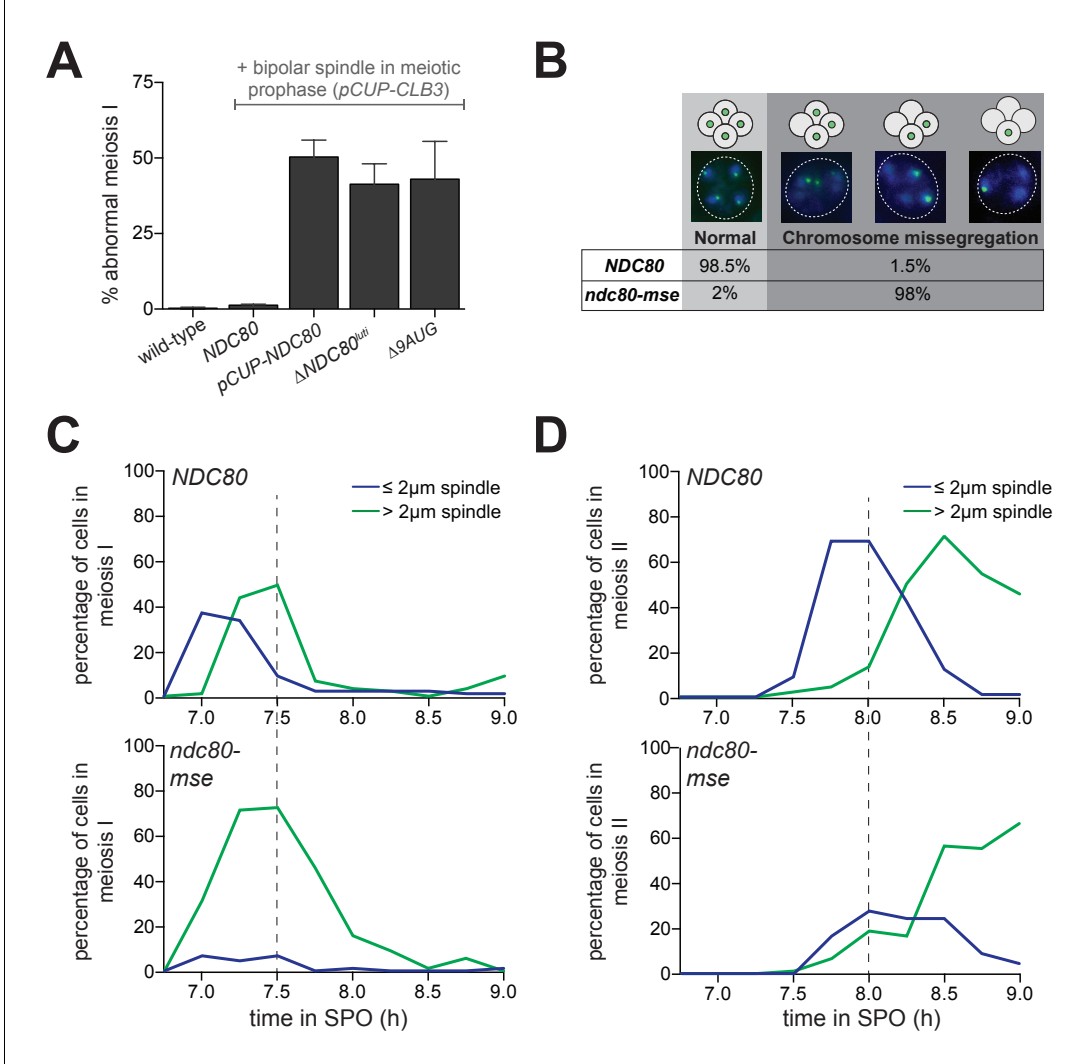

**Figure 7.** Temporal regulation of Ndc80 level by *NDC80^luti* and *NDC80^ORF* in meiosis is required for proper meiotic chromosome segregation. (**A**) Sister chromatid segregation in wild type (UB2942), *pCUP-CLB3* (UB877), *pCUP-CLB3 pCUP-NDC80* (UB880), *pCUP-CLB3 ΔNDC80^luti* (UB2940), and *pCUP-CLB3 Δ9AUG* (UB2936) cells. Cells were induced to sporulate by transferring to SPO, and 6 hr later, expression of the cyclin Clb3 was induced by addition of CuSO₄. Immediately after induction, cells were released from pachytene by addition of β-estradiol. Samples were taken 1 hr 45 min after the release. Premature segregation of sister chromatids in meiosis I (abnormal meiosis I) was detected as two separated GFP dots in binucleates, one in each nucleus. The average fraction of binucleates that displayed sister segregation in meiosis I from three independent experiments, as well as the standard error of the mean, was graphed. 100 cells were counted per strain, per experiment. (**B**) Chromosome segregation accuracy in wild type (UB5876) and *ndc80-mse* (UB5437) strains was determined by counting homozygous CENV-GFP dots in tetranucleates. Samples were taken 7.5 hr after transfer to SPO when most cells had completed meiosis in an asynchronous system. The fraction of tetranucleates that displayed normal segregation (one GFP dot in each nucleus), or missegregation (multiple or zero GFP dots in any of the four nuclei) was quantified. The average fraction of normal segregation or missegregation from two independent experiments is shown. Over 100 cells were counted per strain, per experiment. (**C–D**) Percentage of wild type (UB4074) and *ndc80-mse* (UB3392) cells with meiosis I spindles (shown in C) or meiosis II spindles (shown in D) that were longer than 2 μm, as well as the percentage of cells with spindles that were shorter than 2 μm. Both strains harbor the *pGAL-NDT80 GAL4-ER* system. After 6 hr in SPO, the cells were released from pachytene by addition of β-estradiol, and samples were taken every 15 min after the release. Over 100 cells per time point were quantified, and the results of one representative repeat from two independent experiments are shown.

DOI: https://doi.org/10.7554/eLife.27417.026

The following figure supplement is available for figure 7:

**Figure supplement 1.** Mutation of the MSE site upstream of *NDC80* prevents spore formation.
DOI: https://doi.org/10.7554/eLife.27417.027

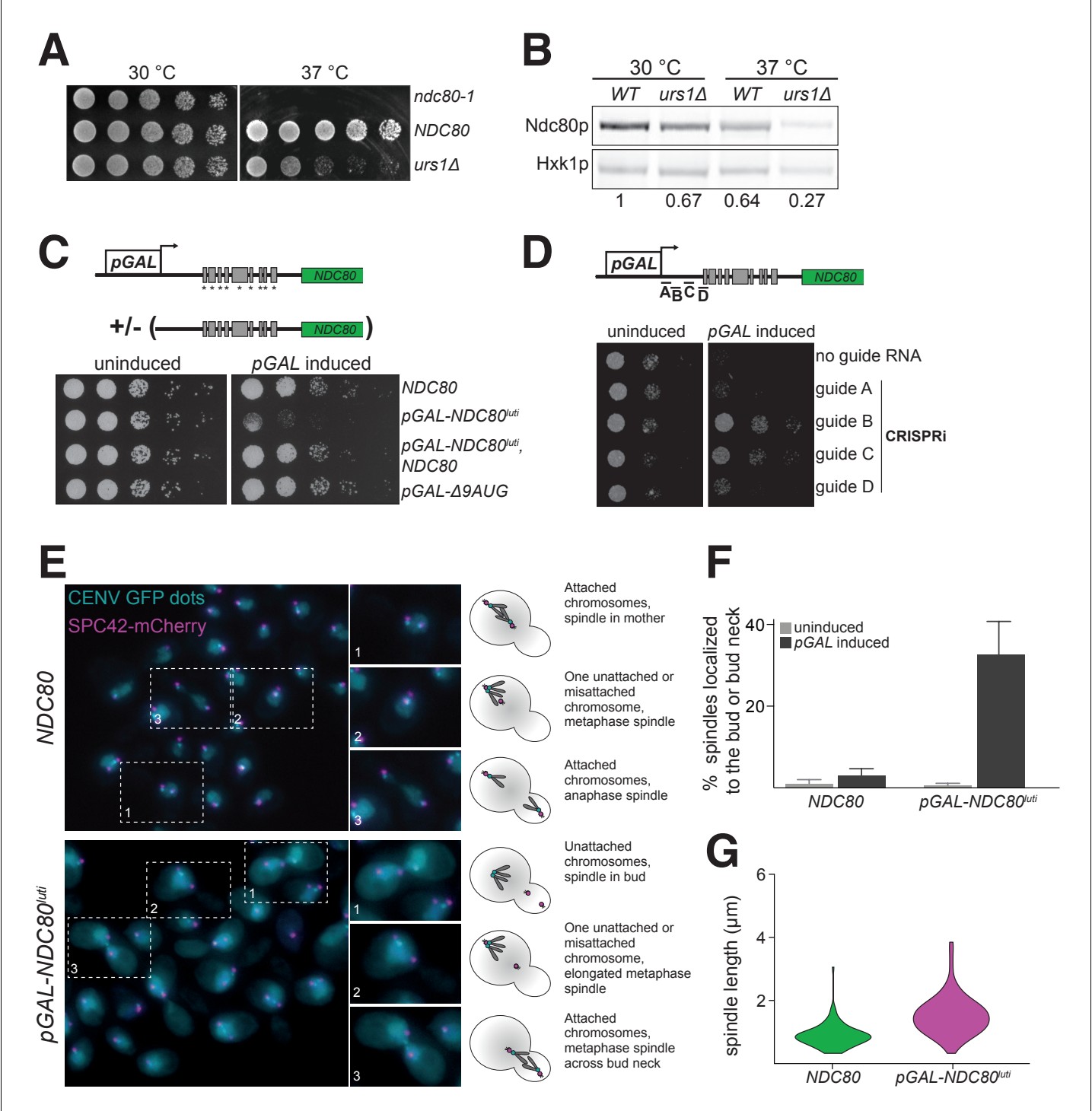

**Figure 8.** Misexpression of *NDC80^luti* outside of meiosis causes severe growth defects due to kinetochore dysfunction. (**A**) Growth phenotype of *ndc80-urs1Δ* cells at 30°C and 37°C. Temperature-sensitive *ndc80-1* (UB494), wild type (UB3262), and *urs1Δ* (UB4212) cells were serially diluted and grown on nutrient rich medium (YPD) plates at 30°C or 37°C for 2 days. (**B**) Ndc80 level in wild type (UB3262) and *urs1Δ* (UB4212) cells grown at 30°C or 37°C. For each condition, equal $OD_{600}$ of cells were taken, and Ndc80 was visualized by anti-V5 immunoblot. Hxk1, loading control. WT, wild type. The number under each lane is the ratio of the relative Ndc80 levels (normalized to Hxk1 levels) compared with that of wild type at 30°C. The results of one representative repeat from two independent experiments are shown. (**C**) Growth phenotype of haploid control (UB1240), *pGAL-NDC80^luti* (UB1217), *pGAL-NDC80^luti* with a second copy of *NDC80* at the *LEU2* locus (UB8001), and *pGAL-Δ9AUG* (UB1323). Cells were serially diluted and grown on YEP-raffinose/galactose (YEP-RG) plates (uninduced) or YEP-RG plates supplemented with β-estradiol (*pGAL* induced) at 30°C for 2 days. (**D**) Growth phenotype of the *pGAL-NDC80^luti* cells carrying a *pGAL*-inducible dCas9-Mxi1 and a vector for one of the following guide RNAs: gRNA A (UB6297),
*Figure 8 continued on next page*

*Figure 8 continued*

gRNA B (UB6299), gRNA C (UB6301), or gRNA D (UB6302). The control strain (UB6295) carries an empty vector. Cells were serially diluted and grown on SC – leu raffinose + galactose plates (uninduced) or SC – leu raffinose + galactose plates supplemented with β-estradiol (*pGAL* induced) at 30°C for 2 days. (E), (F), and (G) Phenotypic characterization of cells expressing *pGAL-NDC80^luti^*. Both the control (UB8682) and *pGAL-NDC80^luti^* (UB8684) cells harbor homozygous CENV-GFP dots and Spc42-mCherry (spindle pole body marker). The strains were grown overnight in YEP-RG, and samples were collected at 0 hr and 6 hr after *pGAL* induction by β-estradiol. (E) Representative images of wild type cells and the cells expressing *NDC80^luti^* after 6 hr of *pGAL* induction. Enlarged images of the boxed regions are shown in the middle. To the right are schematics of the microtubule-kinetochore attachment status in each class of phenotype observed. (F) Quantification of the spindle localization data shown in (E). Among the cells with separated spindle poles, the percentage of cells that had a spindle shorter than 2 µm and were abnormally localized (i.e. across the bud neck or entirely within the bud) is displayed. 100 cells were counted per strain, for each condition. The average percentage and the standard deviation from three independent experiments are shown. (G) Quantification of the spindle length in cells with at least one chromosome V not attached to a spindle pole body. Data from (E), specifically in cells with either both CENV-GFP dots associated with a single spindle pole or both CENV-GFP dots completely dissociated from either spindle pole. This allows analysis of populations of cells that are either in Sphase/early mitosis (after SPB duplication, but before chromosome alignment) or are unable to properly attach their chromosomes. A representative replicate out of three independent experiments was graphed as a violin plot. 100 cells were analyzed per strain, per replicate.

DOI: https://doi.org/10.7554/eLife.27417.028

The following figure supplement is available for figure 8:

**Figure supplement 1.** The distribution of spindle lengths in mitotic cells expressing *NDC80^luti^* indicates that cells arrest in metaphase.

DOI: https://doi.org/10.7554/eLife.27417.029

## Discussion

In this study, we have identified an integrated regulatory circuit that controls the inactivation and subsequent reactivation of the meiotic kinetochore (*Figure 9*). This circuit controls the synthesis of a limiting kinetochore subunit, Ndc80, and relies on the regulated expression of two distinct *NDC80* mRNAs. A meiosis-specific switch in promoter usage induces the expression of a 5′ extended transcript isoform, *NDC80^luti^*, which itself cannot produce Ndc80 protein. Rather, its function is purely regulatory. Transcription of this alternate isoform leads to repression of the protein-translating *NDC80^ORF^* isoform in *cis*. This results in inhibition of Ndc80 protein synthesis and ultimately the inactivation of kinetochore function in meiotic prophase. Reactivation of the kinetochore is achieved by the transcription of *NDC80^ORF^* upon exiting meiotic prophase. Temporally coordinated by two master transcription factors, the timely expression of these two mRNA isoforms is essential for kinetochore function, accurate chromosome segregation, and gamete viability. Altogether, our study describes a new gene regulatory mechanism and provides insight into its biological purpose.

### A limiting subunit controls kinetochore function in meiosis

In meiosis, kinetochore function is transiently inactivated to facilitate accurate chromosome segregation (*Miller et al., 2013*). This transient inactivation is achieved by the removal of the outer kinetochore from chromosomes and has been described in organisms ranging from yeast to mice (*Asakawa et al., 2005*; *Kim et al., 2013*; *Meyer et al., 2015*; *Miller et al., 2012*; *Sun et al., 2011*). In budding yeast, we found that outer kinetochore removal is mediated by limiting the abundance of a single subunit, Ndc80. Ndc80 is the only member of its complex whose protein abundance is essentially absent in meiotic prophase (*Meyer et al., 2015* and *Figure 1C*). Furthermore, prophase overexpression of *NDC80*, but none of the other Ndc80 complex subunits, promotes premature spindle attachments and causes meiotic chromosome segregation errors (*Figure 1D*). Thus, in the case of the meiotic kinetochore, the cell regulates the activity of a multi-protein complex by limiting the availability of a single subunit.

The control of protein complex activity through the limitation of a key subunit is a more general principle. A genome-wide study that analyzed the composition of protein complexes during the cell cycle revealed that in budding yeast, most protein complexes have both constitutively and periodically expressed subunits (*de Lichtenberg et al., 2005*). It is proposed that due to the periodically expressed subunits, these protein complexes assemble "just-in-time" to restrict their function to specific cell cycle stages (*de Lichtenberg et al., 2005*). The luti-mRNA-dependent regulatory circuit described here may more broadly address how regulated subunits are provided "just-in-time" and, importantly, at no other time.

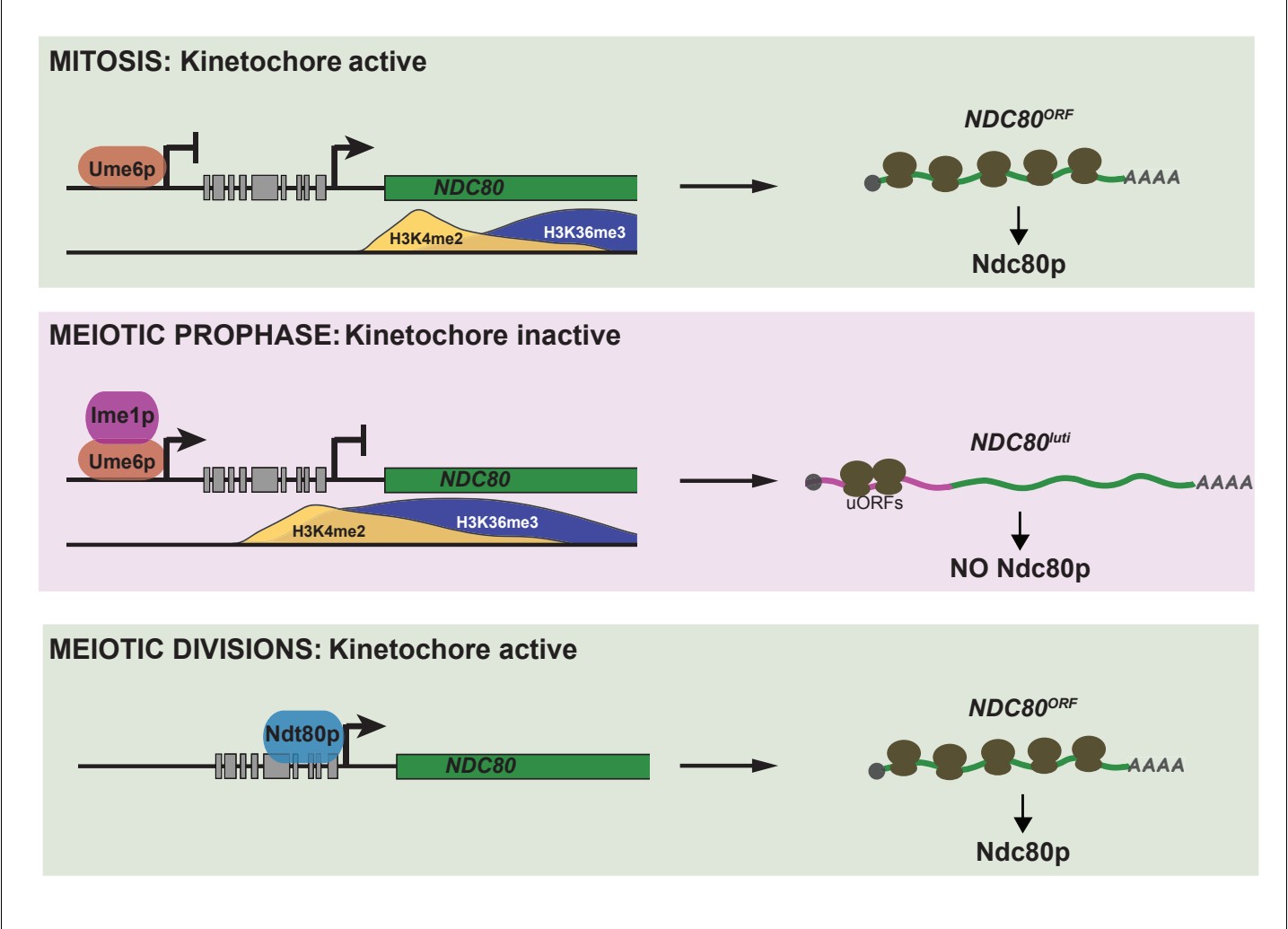

**Figure 9.** Model of *NDC80* gene regulation in budding yeast. During vegetative growth, a stage in which kinetochores are active, a short *NDC80* mRNA isoform *NDC80^ORF* is expressed, and the 5' extended isoform *NDC80^luti* is repressed by Ume6. Translation of *NDC80^ORF* results in Ndc80 protein synthesis (top panel). At meiotic entry, the master transcription factor Ime1 induces expression of *NDC80^luti*. Transcription from this distal *NDC80^luti* promoter silences the proximal *NDC80^ORF* promoter through a mechanism that increases H3K4me2 and H3K36me3 marks over the *NDC80^ORF* promoter (See the accompanying paper Chia et al., for details). *NDC80^luti* does not support Ndc80 synthesis due to translation of the uORFs. The overall synthesis of Ndc80 is repressed in meiotic prophase, and the kinetochores are inactive (middle panel). As cells enter the meiotic divisions, the transcription factor Ndt80 induces *NDC80^ORF* re-expression, allowing for Ndc80 re-synthesis and formation of active kinetochores (bottom panel).

DOI: https://doi.org/10.7554/eLife.27417.030

The following figure supplements are available for figure 9:

**Figure supplement 1.** Clustal analysis for the upstream intergenic region of the *NDC80* locus across five *Saccharomyces* species.
DOI: https://doi.org/10.7554/eLife.27417.031

**Figure supplement 2.** Clustal analysis for the upstream intergenic region of the *NDC80* locus across five *Saccharomyces* species.
DOI: https://doi.org/10.7554/eLife.27417.032

## *NDC80^luti* is an mRNA that does not produce protein

A key aspect of the work presented here is the surprising finding that an mRNA can serve a purely regulatory function. Indeed, *NDC80^luti* is a bona fide mRNA. It is poly-adenylated, is engaged by the ribosome and, most importantly, when the uORF start codons are ablated, Ndc80 protein is translated from this extended mRNA isoform (*Brar et al., 2012* and *Figure 3*). Moreover, *NDC80^luti* is likely a RNA Polymerase II transcript because its promoter is occupied by the pre-initiation complex

member Sua7 (TFIIB) and because Pol II-associated chromatin marks are detected downstream of the *NDC80^luti* promoter when this transcript is made (Chia et al., accompanying manuscript). *NDC80^luti* cannot be decoded by the ribosome due to the presence of AUG-uORFs contained in its extended 5'leader. By competitively engaging the ribosome, these uORFs prevent translation of Ndc80 protein. The polypeptides that the uORFs encode are unlikely to play a role in the repression of kinetochore function as the uORFs can be minimized to 2-codon units while maintaining *NDC80^luti*-based repression (*Figure 3*). Interestingly, upstream AUG codons are also present in the putative *NDC80^luti* mRNAs predicted from the other fungal species. Three regions were enriched for the presence of such AUGs (*Figure 9—figure supplements 1* and *2*), but the sequences and the length of these putative uORFs did not seem to be conserved (*Supplementary file 1G*). This observation is consistent with the idea that the act of uORF translation, rather than the identity of the uORF peptides, serves as a conserved feature in evolution.

The repressive nature of the uORFs contained in *NDC80^luti* mirrors those found in the uORF-containing prototype transcript, *GCN4* (*Mueller and Hinnebusch, 1986*). However, in the case of *GCN4*, changes in nutrient availability can relieve the uORF-mediated translational repression, whereas for *NDC80^luti*, the uORF-mediated repression appears to be constitutive. In both cases, *GCN4* and *NDC80* can exist in on and off states. For *GCN4*, this switch is manifested in the two translational states of the same mRNA molecule. For *NDC80*, the switch is manifested instead by two distinct transcripts, one, which results in protein synthesis and one, which represses protein synthesis. It is important to note that for other potential luti-mRNAs, the precise mechanism of translational repression may not be conserved and could instead involve other means such as RNA hairpins or binding sites for translational repressors.

## The function of *NDC80^luti* mRNA is purely regulatory

Why do meiotic cells express an mRNA that does not encode any functional polypeptides? We propose that the biological purpose of *NDC80^luti* is to shut down Ndc80 protein synthesis by repressing *NDC80^ORF* in *cis*, thereby inactivating kinetochore function during meiotic prophase. Multiple lines of evidence support this model. First, disruption of *NDC80^luti* expression in meiosis results in elevated levels of *NDC80^ORF* and Ndc80 protein in meiotic prophase, leading to premature kinetochore activation (this study and Chia et al., accompanying manuscript). Second, induction of *NDC80^luti* transcription in *cis* is sufficient to repress *NDC80^ORF* and inactivate kinetochore function in mitotic cells (this study). Third, transcription of *NDC80^luti* introduces repressive chromatin marks at the *NDC80^ORF* promoter that are necessary for the downregulation of *NDC80^ORF* and Ndc80 protein (Chia et al., accompanying manuscript). Altogether, these findings strongly suggest that the primary function of the *NDC80^luti* mRNA is to turn off the *NDC80* gene.

It is important to note that our study only addresses the mechanism of how Ndc80 protein synthesis is repressed in meiotic prophase. Indeed, efficient and timely reduction of Ndc80 protein levels may require regulated proteolytic mechanisms not yet elucidated. Further studies are necessary to determine if proteolysis plays a role in the rapid removal of the outer kinetochore in meiotic prophase and if so, by what means this proteolysis is achieved.

## Transcription factor-driven gene repression by luti-mRNA: an evolutionary perspective

Why do budding yeast cells use this seemingly complex mechanism, which relies on the transcription of an undecoded mRNA isoform, to repress a kinetochore gene during meiosis? We would argue from an evolutionary point of view that this solution could be both economical and highly flexible. First, the meiotic cell is co-opting two existing transcription factors, Ime1 and Ndt80, for roles in activating *and* repressing gene expression, obviating the need to evolve novel trans-acting factors. This mechanism also ensures temporal coordination of gene activation *and* inactivation using the same transcription factor. In the case of *NDC80*, the luti-mRNA rides the Ime1 wave of gene expression to shutoff kinetochore function while the protein-coding mRNA rides the subsequent Ndt80 wave to reactivate the kinetochore for the division phases. While transcription factors have previously been implicated in the repression of downstream promoters (*Bird et al., 2006*; *Martens et al., 2004*; *Shearwin et al., 2005*; *van Werven et al., 2012*), our study is the first clear demonstration

that it is the choice of promoter and the identity of the resulting mRNA isoform that governs whether a gene is turned on or turned off by a given transcription factor.

This mode of gene repression relies on two sets of *cis*-regulatory sequences, which are evolutionarily flexible (*Carroll, 2008*; *Stern and Orgogozo, 2008*; *Wittkopp and Kalay, 2011*). The first *cis*-acting sequence is the distal transcription factor-binding site, which induces transcription of *NDC80^{luti}*, and, in concert with co-transcriptional chromatin modifications, silences the downstream canonical promoter activity. The second *cis*-acting sequence is the AUG-uORFs within the extended 5' leader of the luti-mRNA, which prevents downstream ORF translation. Inherent to a mechanism that is so heavily reliant on *cis*-regulatory elements is the notion that minor changes in the DNA sequence can impact gene expression at a multitude of levels, thus tuning gene output. This tuning can be manifested at the level of nucleosome spacing, strength of transcription factor binding and translational regulation. Therefore, the cell has a vast evolutionary space, which can be explored through small changes in DNA sequence.

## Pervasiveness of luti-mRNA biology in yeast meiosis and beyond

The defining sequence features of the *NDC80* luti-mRNA are a 5'-extended mRNA leader coupled with repressive uORFs contained in this extended leader. Analysis of the mRNA-seq and ribosome profiling datasets of meiotic yeast revealed hundreds of transcripts with potential luti-like signatures (*Brar et al., 2012*). In support of this idea, two other genes, *ORC1* and *BOI1*, have been shown to express meiosis-specific transcript isoforms with uORF-containing leader extensions (*Xie et al., 2016* and *Liu et al., 2015*). Rather than dissecting each candidate luti-mRNA on a case by case basis, future studies that integrate additional genome-wide datasets to measure stage-specific transcription factor binding sites, transcription-coupled chromatin modification states, mRNA translation status with isoform specificity and protein abundance would result in a high-confidence map of luti-mRNAs and aid in the dissection of their cellular functions.

Beyond budding yeast meiosis, can the regulatory circuit described in our study be present in other developmental programs and in other organisms? We would argue so, because various organisms also possess the three principles of this module, namely, alternative promoter usage, transcription-coupled repression, and uORF-mediated translational repression. Alternative promoter usage is widespread in development and among different cell types. For example, in the fruit fly, more than 40% of developmentally expressed genes have at least two promoters with distinct regulatory programs (*Batut et al., 2013*). Half of human genes have more than one promoter, resulting in the expression of mRNA isoforms with 5' heterogeneity (*Kimura et al., 2006*). Furthermore, transcription-based interference mechanisms, as well as transcription-coupled histone modifications, have been described in a variety of organisms (*Corbin and Maniatis, 1989*; *Eissenberg and Shilatifard, 2010*; *Shearwin et al., 2005*; *Wagner and Carpenter, 2012*). Finally, recent studies have shown that uORF translation is much more widespread than traditionally believed and acts in a regulatory manner (*Calvo et al., 2009*; *Chew et al., 2016*; *Johnstone et al., 2016*). Therefore, we envision that the regulatory circuit described here can be used as a roadmap in future studies to uncover transcription-coupled gene repression during cell fate transitions across multiple species.

## Interpreting genome-wide data in the context of luti-mRNA biology

A key implication of this model of gene regulation is a blurring of the line between "coding" and "non-coding" RNAs. Seminal work has uncovered multiple classes of non-coding RNAs that play regulatory functions in the cell, such as long non-coding RNAs, microRNAs, small interfering RNAs, and piwiRNAs (*Ambros, 2001*; *Batista and Chang, 2013*; *Cech and Steitz, 2014*; *Guttman et al., 2009*). Our study demonstrates that mRNAs, which are deemed protein coding units, can themselves be direct regulators of gene expression by at least two simultaneous means: they can induce transcription-coupled silencing of a downstream promoter, and features in their 5' leaders, such as the presence of uORFs or secondary structures, could directly impact translation efficiency in a positive or negative manner (*Arribere and Gilbert, 2013*; *Brar et al., 2012*; *Rojas-Duran and Gilbert, 2012*). Notably, multiple studies have reported poor correlation between mRNA and protein abundance (*Maier et al., 2009*). For those mRNAs that anti-correlate with their protein levels, this apparent contradiction might be due to a luti-mRNA being misattributed as a canonical protein-coding transcript. Our study could dramatically transform the way we understand the function of alternate

mRNA isoforms and aid in the proper biological interpretation of genome-wide transcription studies.

## Materials and methods

### Yeast strains and plasmids

All the strains used in this study are described in *Supplementary file 1A* and are derivatives of SK1. The *pGAL-NDT80 GAL4-ER* and the *pCUP-IME1 pCUP-IME4* synchronization systems have been described previously (*Benjamin et al., 2003*; *Berchowitz et al., 2013*). The centromeric TetR/TetO GFP dot assay is described in (*Michaelis et al., 1997*). The *ndc80-1* temperature-sensitive mutant was first described in (*Wigge et al., 1998*), the Zip1::GFP (700) described in (*Scherthan et al., 2007*), and *pCUP-NDC80 pCUP-CLB3* described in (*Miller et al., 2012*). NDC80-3V5, NUF2-3V5, SPC24-3V5, SPC25-3V5, pCUP-NUF2, pCUP-SPC24, pCUP-SPC25, pGAL-NDC80$^{luti}$, pGAL-$\Delta$9AUG, ndc80$\Delta$, nuf2$\Delta$, ($\Delta-600$ to $-300$)-NDC80, and ($\Delta-600$ to $-400$)-NDC80 were generated at the endogenous gene loci using PCR-based methods (*Longtine et al., 1998*). The V5 plasmid is kind gift from Vincent Guacci. Primer sequences used for strain construction can be found in *Supplementary file 1B*. Single integration plasmids carrying either *NDC80* or *NUF2* were constructed by Gibson Assembly (*Gibson et al., 2009*), and were digested with PmeI to integrate at the *LEU2* locus. For *NDC80*, the *LEU2* integration plasmid included the SK1 genomic sequence spanning from 1000 bp upstream to 357 bp downstream of the *NDC80* coding region; and for *NUF2*, spanning from 1000 bp upstream to 473 bp downstream of the *NUF2* coding region. Both constructs included a C-terminal fusion of the 3V5 epitope to *NDC80* and *NUF2*, and both completely rescued the full deletion of *NDC80* or *NUF2*, respectively. Deletions (*ndc80-urs1$\Delta$* and ($\Delta-600$ to $-479$)-NDC80) and point mutations (*ndc80-mse*) were generated from the *NDC80 LEU2* single integration plasmid using the site-directed mutagenesis kit (Q5 Site-Directed Mutagenesis Kit, *NEB, Ipswitch, MA*). The entire URS1 site and the "A" right upstream of the site were deleted in the *ndc80-urs1$\Delta$* strain. The *ndc80-mse* construct has two C to A mutations, marked using black diamonds in *Figure 6B*. The $\Delta$6AUG, $\Delta$9AUG, *mini uORF*, NDC80$^{luti-Ter}$, and NDC80$^{luti}$-NUF2 constructs were generated by Gibson assembly (*Gibson et al., 2009*) using the *NDC80* and *NUF2 LEU2* integration plasmids, as well as gBlocks gene fragments (IDT, *Redwood City, CA*) for the $\Delta$9AUG and *mini uORF* constructs. *SNR52* promoter-controlled guide RNAs targeting NDC80$^{luti}$ (A-D) were cloned into a 2-micron plasmid carrying a *LEU2* selectable marker (pRS425 backbone). See *Supplementary file 1C* for the full list of the integration and 2-micron plasmids.

### *pCUP-IME1 pCUP-IME4* synchronous sporulation

Synchronously sporulating cell cultures were prepared as in (*Berchowitz et al., 2013*). In short, the endogenous promoters of *IME1* and *IME4* were replaced with the inducible *CUP1* promoter. Diploid cells were grown in YPD (1% yeast extract, 2% peptone, 2% glucose, and supplemented with 22.4 mg/L uracil and 80 mg/L tryptophan) for 20–24 hr at room temperature. For optimal aeration, the total volume of the flask exceeded the volume of the medium by 10 fold. Subsequently, cells were transferred to BYTA (1% yeast extract, 2% bacto tryptone, 1% potassium acetate, 50 mM potassium phthalate) and grown for another 16–18 hr at 30°C. The cells were then pelleted, washed with sterile milliQ water, and resuspended at 1.85 OD$_{600}$ in sporulation (SPO) media (0.5% (w/v) potassium acetate [pH 7], 0.02% (w/v) raffinose) at 30°C. To initiate synchronous sporulation, expression of *IME1* and *IME4* was induced 2 hr after cells were transferred to SPO by adding copper (II) sulphate to a final concentration of 50 μM.

### *pGAL-NDT80* synchronous meiotic divisions

The *pGAL-NDT80 GAL4-ER* system was used to generate populations of cells synchronously undergoing the meiotic divisions (Carlile and Amon, 2008). Cells were prepared for meiosis as in the *pCUP-IME1 pCUP-IME4* protocol, and resuspended at 1.85 OD$_{600}$ in SPO. The flasks were placed at 30°C for 5–8 hr to block cells in meiotic prophase (See figure legend for the specific arrest duration for each experiment). To release cells from pachytene, *NDT80* expression was induced with 1 μM β-estradiol. Subsequently, cells progressed through meiosis synchronously.

## Alpha-factor arrest-release mitotic time course

*MAT***a** cells were first grown to an OD$_{600}$ of 1–2 at 30°C in YPD, diluted back to OD$_{600}$ 0.005 in YEP-RG (2% raffinose and 2% galactose in YEP supplemented with 22.4 mg/L uracil and 80 mg/L tryptophan), and then grown at room temperature for 15–17 hr. Exponentially growing cells were diluted again to an OD$_{600}$ of 0.19 in YEP-RG, and arrested in G1 with 4.15 µg/mL alpha-factor, and 1.5 hr later, an additional 2.05 µg/mL of alpha-factor was added to the cells. After 2 hr in alpha-factor, 1 µM β-estradiol was added to cultures to induce *pGAL* expression. One hour after the β-estradiol addition, cells were filtered, rinsed with YEP (10 times volume of the culture volume) to remove the alpha-factor, and placed into a receiving flask containing YEP-RG with 1 µM β-estradiol. Time points were taken before β-estradiol induction, before release, and every 15 min after release, for 3 hr.

## Conservation analysis

Clustal analysis (*Goujon et al., 2010*; *Sievers et al., 2011*) was performed using the genomic sequences of *S. bayanus*, *S. kudriavzevii*, *S. mikatae*, *S. cerevisiae* and *S. paradoxus* from *Saccharomyces sensu stricto* genus (*Scannell et al., 2011*), and imported into the Webpage of the Clustal Omega Multiple Sequence Alignment tool < http://www.ebi.ac.uk/Tools/msa/clustalo/ >.

## Chromatin immunoprecipitation

The Ume6-3V5 chromatin immunoprecipitation experiments were performed as described previously with the following modifications (*van Werven et al., 2012*). Cells were fixed with formaldehyde (1% v/v) for 15 min. Frozen cell pellets were disrupted 4 times (5 min each) using a Beadbeater (Mini-Beadbeater-96, *Biospec Products*, *Bartlesville*, *OK*). Chromatin was sheared 5 × 30 s ON/30 s OFF with a Bioruptor Pico (*Diagenode*, *Denville*, *NJ*) to a fragment size of ~200 bp. Chromatin extracts were incubated with 20 µL of anti-V5 agarose beads (A7345, *Sigma*, *St. Louis*, *MO*) at 4°C. The Ndt80-3V5 chromatin immunoprecipitation experiments were performed as described previously with the same modifications as used for Ume6-3V5 except for the sonication conditions (*Strahl-Bolsinger et al., 1997*). Chromatin was sheared 5 × 10 s ON/30 s OFF with a Bioruptor Pico (*Diagenode*) to a fragment size of ~500 bp. Reverse crosslinked input DNA and immunoprecipitated DNA fragments were amplified with Absolute SYBR green (AB4163/A, *Thermo Fisher*, *Waltham*, *MA*) and quantified with a 7500 Fast Real-Time PCR machine (*Thermo Fisher*) using the primer pairs directed against the upstream region and the coding region of *NDC80*, the *MAM1* promoter, and the *IME2* promoter. We also measured the signals from the *NUF2* promoter and *HMR*, regions that do not display significant binding for either of the transcription factors. The oligonucleotide sequences used are listed in *Supplementary file 1D*.

## Fluorescence microscopy (CENV-GFP dots and Spc42-mCherry)

Cells were fixed with 3.7% formaldehyde at room temperature for 15 min, washed once with potassium phosphate/sorbitol buffer (100 mM potassium phosphate [pH 7.5], 1.2 M sorbitol), and then permeabilized with 1% Triton X-100 with 0.05 µg/mL DAPI in potassium phosphate/sorbitol buffer. Cells were imaged using a DeltaVision microscope with a 100x/1.40 oil-immersion objective (DeltaVision, *GE Healthcare*, *Sunnyvale*, *CA*) and filters: DAPI (EX390/18, EM435/48), GFP/FITC (EX475/28, EM525/48), and mCherry (EX575/25, EM625/45). Images were acquired using the softWoRx software (softWoRx, *GE Healthcare*).

## Quantification of spindle length and CENV-GFP dots in mitosis

For *Figure 8E–G* and *Figure 8—figure supplement 1*, diploid cells were first grown to an OD$_{600}$ of 1–2 at 30°C in YPD. They were then diluted to an OD$_{600}$ of 0.002 in YEP-RG and grown at 30°C for 16 hr. Exponentially growing cells were diluted back to an OD$_{600}$ of 0.2 in YEP-RG and induced to express *NDC80$^{luti}$* with 1 µM β-estradiol. Samples were taken before induction and 6 hr after induction. Images were acquired as described in the fluorescence microscopy method section, and analysed using the FIJI image processing software (RRID:SCR_002285, *Schindelin et al., 2012*). First, maximum-intensity projection was performed. Second, projected spindle length (defined as the distance between Spc42-mCherry foci) was measured using the "measure" plugin. The distribution of the projected spindle length was graphed as violin plots using (BoxPLotR RRID:SCR_015629, *Spitzer et al., 2014*). Third, in cells with separated spindle poles, the status of the Spc42-mCherry

association with CENV-GFP dots was categorized as 1) each Spc42-mCherry focus is associated with a CENV-GFP dot, 2) only one Spc42-mCherry focus is associated with CENV-GFP dots (either one or both of the GFP dots), or 3) neither Spc42-mCherry focus is associated with a CENV-GFP dot. After categorizing the localization of the CENV-GFP dots, the projected spindle length was measured for spindles in category 2 and 3, and the spindle length distributions were graphed as violin plots using (BoxPLotR RRID:SCR_015629, *Spitzer et al., 2014*). Finally, in cells with separated spindle poles, the location of the spindle was recorded as 1) in the mother, 2) across the bud neck, or 3) in the bud. The percentage of spindles that were both less than 2.0 μm and abnormally localized (across the bud neck or in the bud) was calculated. For each analysis, 100 cells were counted.

## Indirect immunofluorescence

Tubulin indirect immunofluorescence was performed as described (*Kilmartin and Adams, 1984*) using a rat anti-tubulin antibody (MCA78G, *Bio-rad Antibodies, Kidlington, UK*) at a dilution of 1:200 and a pre-absorbed anti-rat FITC antibody (712-095-153, *Jackson ImmunoResearch Laboratories, Inc. West Grove, PA*) at a dilution of 1:200. The meiotic stage of a cell was determined based on its spindle and DAPI morphologies. Metaphase I spindles were defined as a short bipolar spindle spanning a single DAPI mass; an anaphase I spindle was defined as a single elongated spindle spanning two DAPI masses; a pair of metaphase II spindles were defined as two short bipolar spindles each spanning a distinct DAPI mass within a single cell; and finally, a pair of anaphase II spindles was defined as two elongated spindles with 4 DAPI masses within a single cell. To image spindle samples for characterization of spindle length, z stacks (8–10 slices) were acquired with a step size of 0.5 μm using the DeltaVision microscope (*GE Healthcare*) described in the fluorescence microscopy section. To measure the projected spindle length, maximum-intensity projection of these images was generated by FIJI (RRID:SCR_002285, *Schindelin et al., 2012*). Next, the projected spindle length (defined as the spindle pole-to-pole distance) was measured using the "measure" plugin (*Schindelin et al., 2012*), and cells were staged to be in either meiosis I or meiosis II depending on the number of bipolar spindles. For cells undergoing meiosis II, both spindles were quantified, but only the longer of the two was reported. For each time point, the percentage of cells in each category was quantified: 1) meiosis I spindles that were less than 2 μm, 2) meiosis I spindles that were over 2 μm, 3) meiosis II spindles that were less than 2 μm, and 4) meiosis II spindles that were over 2 μm. Over 100 cells per time point were quantified.

## Northern blotting

A previously described northern blot protocol was modified as below (*Koster et al., 2014*). RNA was extracted with acid phenol:chloroform:isoamyl alcohol (125:24:1; pH 4.7) and then isopropanol precipitated. RNA samples (8–10 μg) were denatured in a glyoxal/DMSO mix (1 M deionized glyoxal, 50% v/v DMSO, 10 mM sodium phosphate buffer pH 6.5–6.8) at 70°C for 10 min and then separated on a 1.1% agarose gel for 3 hr at 80 V. RNAs were transferred onto nylon membranes overnight by capillary transfer. The membranes were blocked for at least 3 hr at 42°C in ULTRAhyb Ultrasensitive Hybridization Buffer (*Thermo Fisher*) before hybridization. Radioactive probes were synthesized using a Prime-It II Random Primer Labelling Kit (*Agilent, Santa Clara, CA*). The oligonucleotide sequences of the primers used to amplify the *NDC80, NUF2, SCR1,* and *CIT1* DNA template are displayed in *Supplementary file 1D*.

Quantification was performed with FIJI (RRID:SCR_002285, *Schindelin et al., 2012*). For all the images, the LUT (lookup table) was inverted. Then, a rectangular box was drawn around a band of interest. The mean signal intensity (gray-scale) within the box area was calculated using the "measure" plugin. For background subtraction, the same box was moved directly above and below the band, the signal intensity of these two regions was measured, and the average background intensity (top and bottom) was calculated. After subtracting the average background intensity of a given lane from the signal intensity of the band in that lane, this corrected value for each time point was then normalized to the initial time point. The same-sized box was used for all the time points in one experiment.

## Single-molecule RNA FISH

Single-molecule RNA FISH was performed as described (*Raj et al., 2008*) with modifications. All the probes (*Supplementary file 1E* for probe sequences) were designed, synthesized, and labelled by Stellaris (*Biosearch Technologies, Novato, CA*). The unique region of $NDC80^{luti}$ was targeted by twenty 20-mer oligonucleotide probes coupled to CAL Fluor Red 590. Thirty 20-mer probes, coupled to Quasar 670 dye, were targeted to the coding region of *NDC80*. To measure our detection quality, 54 alternating probes (odd and even probes, 27 probes in each set) were designed to target the common region of $NDC80^{luti}$ and $NDC80^{ORF}$, and coupled with Quasar 670 dye and CAL Fluor Red 590 dye, respectively.

For meiosis experiments, cells were sporulated as described above. To fix cells, 160 µL of 37% formaldehyde was added into 1840 µL of meiotic cultures and incubated at room temperature for 20 min with gentle agitation. The fixed samples were moved to 4°C to continue fixing overnight. For vegetative samples, cells were grown in YPD to an $OD_{600}$ of 0.4–0.6, fixed in formaldehyde at room temperature for 20 min, and then prepared for digestion as below.

Cells were washed three times in 1.5 mL cold Buffer B (0.1 M potassium phosphate [pH 7.5], 1.2 M sorbitol) and resuspended in 425 µl digestion buffer (425 µL Buffer B mixed with 40 µL 200 mM Vanadyl ribonucleoside complex (VRC) (*NEB*) with 50 µg of zymolyse (zymolase 100T, *MP Biomedicals, Santa Ana, CA*). Cells were digested at 30°C until approximately 70% of cells were digested. This took about 15–20 min for early meiotic and vegetative samples and 30–35 min for pachytene and post meiotic prophase samples. Digested cells were gently washed with 1 mL of cold Buffer B and resuspended in 1 mL of 70% EtOH for 3.5–5 hr to allow permeabilization. To prepare for hybridization, cells were first incubated in 1 mL of 10% formamide wash buffer (10% formamide, 2X SSC) at room temperature for at least 15 min. For hybridization, each probe set (to a final concentration of 500 nM) and 20 mM VRC were added to hybridization buffer (1% Dextran sulfate (*EMD Millipore, Billerica, MA*), 1 mg/mL *E. coli* tRNA (*Sigma*), 2 mM VRC, 0.2 mg/mL BSA, 1X SSC, 10% formamide (*Thermo Fisher*) in nuclease-free water). Hybridization was performed overnight at 30°C with gentle agitation. Samples were then incubated in the dark for 30 min at 30°C in 1 mL of 10% formamide wash buffer, the buffer was then washed away, cells were stained with DAPI, and resuspended in 50 µL of glucose-oxygen-scavenging buffer (GLOX buffer (10 mM Tris [pH 8.0], 2x SSC, 0.4% glucose)) solution without enzymes. Prior to imaging, 15 µL of GLOX solution with enzyme (1% v/v catalase, 1% v/v glucose oxidase (*Sigma*), 2 mM Trolox (*Sigma*)) was added to the sample. Images were acquired with the DeltaVision microscope (*GE Healthcare*) as described in the fluorescence microscopy section with two additional filters: TRITC (EX542/27, EM597/45) for CAL Fluor Red 590 and CY5 (EX632/22, EM679/34) for Quasar 670. Series of z-stacks (15–25 slices) were acquired with a step size of 0.2 µm.

To quantify FISH spots, maximum-intensity projection of the z-stacks was first generated in (RRID: SCR_002285, *Schindelin et al., 2012*), different channels were split, and these processed images were analysed with custom software written in Matlab (*McSwiggen, 2017*)(*Mathworks, Sunnyvale, CA*). Cell boundaries were hand-drawn. The spot detection code first filtered the raw images using an eight pixel Gaussian kernel to remove background signal. Diffraction-limited spots corresponding to single mRNA were detected using an adaptation of the MTT spot-detection algorithm (*Sergé et al., 2008*), using the following detection parameters: NA: 1.4; detection box: 5 pixels; error rate: 0.1; deflation loops: 0. With these detection settings, many low-intensity fluctuations in background fluorescence were detected as spots. To identify bona fide mRNA molecules, we plotted the signal (defined as the integrated value of the pixel intensities) against the signal-to-noise ratio (SNR; defined as the signal divided by the variance of the pixel values around the detected spot), identified a population of detections that were well separated from the background detections, and chose these signal and SNR values as thresholds. To confirm these threshold choices, we plotted the number of spots detected as a function of the threshold chosen, and found that these thresholds fell within a 'plateau', as others have described (*Senecal et al., 2014*; *Raj et al., 2006*), where an increase in the choice of threshold has little effect on the total number of mRNA detected. Inspection of detected mRNAs, post-threshold, was in good agreement with spots that were manually counted. Once chosen, the same "signal" and "SNR" thresholds were applied to all the images within a replicate. In general, we found that thresholds between replicates varied only

slightly (For CF 590 probes, signal = 1100–1500 and SNR = 2.5–3; for Q670 probes, signal = 1000–2000 and SNR = 2–3).

After detection, spots between the CF 590 and Q 670 probe sets need to be paired to identify $NDC80^{luti}$ and $NDC80^{ORF}$ transcripts. Pairing was done using the *knnsearch* Matlab function to separately identify the closest CF 590 spot for each detected Q 670 spot, and vice versa. Two spots are only considered paired if they are mutual nearest neighbors. Using this as a criterion for pairing, greater than 95% of spot pairs occurred within 2 pixels of each other, which is well within the expected value given any chromatic and detection artifacts between the two color channels. By comparison, fewer than 10% of unpaired spots had nearest neighbor distance of less than four pixels, showing that the probability of misidentifying a spot pair is low. The number of cells with a given number of $NDC80^{luti}$ or $NDC80^{ORF}$ transcripts per cell was graphed as relative frequency histograms. The largest bin of each histogram was normalized to the same length across all the histograms.

## Statistical analysis of smFISH data

Per-cell statistics of paired spots ($NDC80^{luti}$ mRNA), Q 670-only spots ($NDC80^{ORF}$), and CF 590-only spots (false positives, early terminated transcripts, and degradation products) were collected and pooled between biological replicates. First, to determine whether sufficient data had been collected for a given data set, bootstrap analysis of the data was performed. For 500 iterations, statistics from a single cell was randomly sampled from the data, and the mean and variance calculated. This process was repeated for two cells randomly selected from the data, without replacement; then for three cells randomly selected, etc. until one half of the total data set size was reached. A plot of the mean and standard deviation of paired and unpaired spots shows that the mean is stable and that the change in the variance plateaus at a number far below the number of cells assayed, suggesting that our sample size is sufficiently large (*Figure 2—figure supplement 5*). For each sample, over 95 cells were counted and three independent experiments were performed. Thus, for each data set, we could ensure that enough cells were measured to accurately account for the biological variation intrinsic to the data set. To compare across different strains and conditions, the two-tailed non-parametric Wilcoxon Rank Sum test was applied to the pooled data obtained from three independent experimental repeats. The p-value was determined using the *ranksum* function in Matlab (*Mathworks*).

Explanation about *Figure 6H*: Based on our smFISH statistical analysis, the $NDC80^{ORF}$ transcript level in the *urs1Δ* cells did not differ significantly from that of wild type cells, even though there was a clear difference in the northern blot analysis (*Figure 6I*). We consider the possibility that our smFISH quantification method has a technical limitation when the $NDC80^{luti}$ isoform is highly expressed. Since we identified $NDC80^{ORF}$ based on the presence of the Q 670 signal (both transcripts) and the absence of CF 590 signal ($NDC80^{luti}$ unique probes), a missed localization in the CF 590 channel would cause us to over-estimate the number of $NDC80^{ORF}$. In our control experiments using alternating probes (*Figure 2—figure supplement 4*), we measured that ~ 6% of the Q 670 spots lack colocalizing signal from the CF 590 channel. In conditions where the $NDC80^{luti}$ isoform is expressed to the high level observed in wild type meiotic prophase, we expect to miss ~1 CF 590 spot per cell, which would then be interpreted as an extra $NDC80^{ORF}$ molecule. Since the total number of $NDC80^{luti}$ transcripts between wild type and the *urs1Δ* mutant was quite different during meiotic prophase (the median of $NDC80^{luti}$ transcripts is 15 in wild type, that of *urs1Δ* is merely 5, *Figure 6H*), the number of mRNA being mis-classified as $NDC80^{ORF}$ mRNA would also be higher in wild type cells, due to a missed signal from the CF 590 channel. Therefore, it is possible that we over-estimated the number of $NDC80^{ORF}$ mRNA in the wild type strain. Given these limitations, we propose that the difference in transcript levels between the wild type and *urs1Δ* mutant is too subtle to be detected by our smFISH analysis.

## Spot growth assay

Cells were grown on YPG (2% glycerol + YEP) plates overnight, resuspended in milliQ $H_2O$, and then diluted to an $OD_{600}$ of 0.2. 5-fold serial dilutions were performed, and cells were spotted onto YEP-RG plates with or without supplement of 1 μM β-estradiol. The cells were incubated at 30°C for 1–2 days. For experiments in which dCas9 was used to repress $NDC80^{luti}$, cells were first grown on SC-G -leu (0.67% yeast nitrogen base, 2% glycerol, supplemented with adenine, lysine, tyrosine,

phenylalanine, threonine, uracil, tryptophan, and histidine). Serial dilutions were performed as above and cells were spotted onto SC-RG -leu plates (0.67% yeast nitrogen base, 2% raffinose, 2% galactose, supplemented with adenine, lysine, tyrosine, phenylalanine, threonine, uracil, tryptophan, and histidine) with or without 1 μM β-estradiol.

## Immunoblot

Protein extracts were prepared using a trichloroacetic acid (TCA) extraction protocol. Briefly, ~4 $OD_{600}$ units of cells were treated with 5% trichloroacetic acid for at least 15 min at 4°C. Following an acetone wash, the cell pellet was subsequently dried. The cell pellet was lysed with glass beads in lysis buffer (50 mM Tris–HCl [pH 7.5], 1 mM EDTA, 2.75 mM DTT, protease inhibitor cocktail (cOmplete EDTA-free, *Roche, Basel, Switzerland*) using a Mini-Beadbeater-96 (*Biospec Products*). Next, 3x SDS sample buffer (187.5 mM Tris [pH 6.8], 6% ß-mercaptoethanol, 30% glycerol, 9% SDS, 0.05% bromophenol blue) was added and the cell lysate was boiled for 5 min. Proteins were separated by PAGE using 4–12% Bis-Tris Bolt gels (*Thermo Fisher*) and transferred onto nitrocellulose membranes (0.45 μm, *Bio-rad, Hercules, CA*) using a semi-dry transfer apparatus (Trans-Blot Turbo Transfer System, *Bio-rad*). The membranes were blocked for at least 30 min with Odyssey Blocking Buffer (PBS) (*LI-COR Biosciences, Lincoln, NE*) before incubation overnight at 4°C with a mouse anti-V5 antibody (RRID:AB_2556564, R960-25, *Thermo Fisher*) at a 1:2000 dilution. We monitored Hxk1 levels using a rabbit anti-hexokinase antibody (RRID:AB_2629457, H2035, *US Biological, Salem, MA*) at a 1:10,000 dilution, Pgk1 levels with a 1:10,000 diluted mouse anti-Pgk1 antibody (RRID:AB_2532235, SC7167, Molecular Probes, Carlsbad, CA), and Kar2 levels with a 1:200,000 rabbit anti-Kar2 antibody (provided by Mark Rose). Membranes were washed in PBST (phosphate buffered saline with 0.01% tween-20) and incubated with an anti-mouse secondary antibody conjugated to IRDye 800CW at a 1:15,000 dilution (RRID:AB_621847, 926–32212, *LI-COR Biosciences*) and an anti-rabbit antibody conjugated to IRDye 680RD at a 1:15,000 dilution (RRID:AB_10956166, 926–68071, *LI-COR Biosciences*) to detect the V5 epitope and Hxk1, respectively. Immunoblot images were generated and quantified using the Odyssey system (*LI-COR Biosciences*).

## Software

All code used for the analysis of smFISH images has been made available by the authors in the following code repository: https://gitlab.com/tjian-darzacq-lab/Chen_Tresenrider_et_al_2017 (copy archived at https://github.com/elifesciences-publications/Chen_Tresenrider_et_al_2017).

# Acknowledgements

We thank Xavier Darzacq for help with the smFISH analysis platform, Anne Dodson and Stephanie Heinrich for help with the smFISH experimental setup, Haiyan Huang for help with the statistical analysis of the smFISH data, Gloria Brar, Leon Chan, Barbara Meyer, Christopher Mugler, Michael Rape, Jasper Rine, Frank Uhlmann and all members of the Ünal and Brar labs for experimental suggestions and critiques of this manuscript. This work was supported by funds from the March of Dimes (5-FY15-99), Pew Charitable Trusts (00027344), Damon Runyon Cancer Research Foundation (35-15) and Glenn Foundation to EÜ, funds from the Francis Crick institute, which receives its core funding from Cancer Research UK (FC001203), the UK Medical Research Council (FC001203), and the Wellcome Trust (FC001203), to FVW, an A*STAR scholarship to MC, and a NSF Graduate Research Fellowship Grant No. DGE-1106400 to JC.

## Additional information

### Funding

| Funder | Grant reference number | Author |
|---|---|---|
| March of Dimes Foundation | 5-FY15-99 | Elcin Unal |
| Pew Charitable Trusts | 00027344 | Elcin Unal |
| Glenn Foundation for Medical Research | | Elcin Unal |

| | | |
|---|---|---|
| Francis Crick Institute | | Folkert Jacobus van Werven |
| National Science Foundation | | Jingxun Chen |
| Agency for Science, Technology and Research | | Minghao Chia |
| Damon Runyon Cancer Research Foundation | 35-15 | Elcin Unal |

The funders had no role in study design, data collection and interpretation, or the decision to submit the work for publication.

### Author contributions
Jingxun Chen, Amy Tresenrider, Conceptualization, Data curation, Formal analysis, Investigation, Methodology, Writing—original draft, Writing—review and editing; Minghao Chia, Conceptualization, Formal analysis, Investigation, Methodology, Writing—review and editing; David T McSwiggen, Software, Formal analysis, Investigation, Methodology, Writing—review and editing; Gianpiero Spedale, Victoria Jorgensen, Hanna Liao, Formal analysis, Investigation, Methodology; Folkert Jacobus van Werven, Conceptualization, Data curation, Supervision, Funding acquisition, Investigation, Methodology, Project administration, Writing—review and editing; Elçin Ünal, Conceptualization, Data curation, Formal analysis, Supervision, Funding acquisition, Investigation, Methodology, Writing—original draft, Project administration, Writing—review and editing

### Author ORCIDs
Folkert Jacobus van Werven (iD) http://orcid.org/0000-0002-6685-2084
Elçin Ünal (iD) http://orcid.org/0000-0002-6768-609X

### Decision letter and Author response
Decision letter https://doi.org/10.7554/eLife.27417.045
Author response https://doi.org/10.7554/eLife.27417.046

# Additional files

### Supplementary files
• Supplementary file 1.  (A) Detailed genotypes for the strains used in this study. (B) Primers used for strain construction in this study. (C) Plasmids used for strain construction in this study. (D) Primers used for quantitative PCR and northern blotting in this study. (E) smFISH oligonucleotide probes used in this study. The NDC80$^{ORF}$ (Q 670) probe set consists of a mixture of thirty 20-mer oligonucleotide probes that tile the common region shared between NDC80$^{luti}$ and NDC80$^{ORF}$. Each individual probe is labeled with the Quasar 670 dye. The NDC80Long (CF 590) probe set consists of a mixture of twenty 20-mer oligonucleotide probes that tile the unique 5' region of NDC80$^{luti}$. Each individual probe is labeled with the CAL Fluor Red 590 dye. smFISH oligonucleotide probes used in this study. The NDC80Odd (CF 590) probe set consists of a mixture of twenty-seven 20-mer oligonucleotide probes that tile NDC80$^{ORF}$. Each individual probe is labeled with the CAL Fluor Red 590 dye. The NDC80Even (Q 670) probe set consists of a mixture of twenty-seven 20-mer oligonucleotide probes that tile NDC80$^{ORF}$. Each individual probe is labeled with the Quasar 670 dye. (F) Summarized smFISH results for this study. (G) Predicted peptide sequences for the putative AUG uORFs.
DOI: https://doi.org/10.7554/eLife.27417.033

• Transparent reporting form
DOI: https://doi.org/10.7554/eLife.27417.034

### Major datasets
The following previously published dataset was used:

| Author(s) | Year | Dataset title | Dataset URL | Database, license, and accessibility information |
|---|---|---|---|---|
| Brar GA, Yassour M, Friedman N, Regev A, Ingolia NT, Weissman JS | 2012 | High-resolution view of the yeast meiotic program revealed by ribosome profiling | https://www.ncbi.nlm.nih.gov/geo/query/acc.cgi?acc=GSE34082 | Publicly available at the NCBI Gene Expression Omnibus (accession no. GSE34082) |

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
