## [Decision Letter]

Thank you for submitting your article "Kinetochore inactivation by expression of a repressive mRNA" for consideration by *eLife*. Your article has been reviewed by three peer reviewers, and the evaluation has been overseen by a Reviewing Editor and Jessica Tyler as the Senior Editor. The reviewers have opted to remain anonymous.

The reviewers have discussed the reviews with one another and the Reviewing Editor has drafted this decision to help you prepare a revised submission..

Summary:

In this manuscript, Chen et al. identify a gene expression program in budding yeast that temporally controls the expression of Ndc80, a component of the outer kinetochore. Previous work showed that proper control of meiosis requires temporary removal of the outer kinetochore during early stages of meiosis (prophase). In this study, the authors demonstrate that Ndc80 is the limiting outer kinetochore component, whose levels decrease in prophase and increase in meiosis I. The authors uncover a regulatory mechanism controlling Ndc80 levels in which expression of Ndc80 is inhibited by a long, 5'-extended isoform (termed "luti") of the *NDC80* mRNA that is expressed antagonistically with the coding isoform. Translation of the luti mRNA is inhibited by 9 uORFs. Removing all uORF AUG codons relieved repression of *NDC80* mRNA translation consistent with the notion that the upstream AUG codons capture scanning ribosomes to prevent main ORF translation. Inserting stop codons did not affect regulation, indicating that translation in the leader (and not the encoded peptides of the upstream ORFs) is sufficient to block main ORF translation. Additional experiments establish details of the differential regulation of luti vs. *NDC80* mRNA by Ime1 and Ndt80. Interestingly, the 5' regulatory region of luti can be transferred to a different gene and cause that gene to be down-regulated in prophase. Furthermore, up-regulation of *NDC80^luti^* significantly inhibits growth of mitotic cells and causes defects in chromosome segregation.

Overall, this is a very comprehensive and high quality study that addresses an interesting mode of gene regulation. At this point, the field has numerous examples of *cis*-acting noncoding RNAs regulating the expression of a neighboring gene, first shown for the *SER3* gene in yeast involving a non-coding upstream RNA. What makes this study significant is the interesting link to a critical step in meiosis combined with the switch in promoter utilization, which relies on the temporal involvement of two transcriptional regulators of meiotic genes. Also of note is the finding that the luti mRNA is translationally inactive due to uORFs in the leader, a theme that the authors extend in the Discussion. The conclusions in this paper are supported by convincing data, often acquired through complementary approaches, and a carefully written document.

The analysis is very extensive and the supporting data are comprehensive. There are however a few instances in which an important control was lacking, or quantification of results from replicate experiments should be added to bolster the conclusions. Statistical analysis of data from replicates is frequently lacking.

Essential revisions:

1) Figure 1: verify by western analysis that overexpression of Nuf2, Spc24, and Spc25 has actually occurred.

2) Figure 2: analyze an unregulated mRNA as loading control, as the nuclease-resistant Pol I rRNA transcript is not the best control for establishing equal recovery of unstable Pol II mRNA transcripts. Also for this figure, at early times in prophase, fairly high levels of Ndc80p are present, yet the primary transcript is clearly the luti mRNA. What is the source of this pool of Ndc80? Do the authors think it is persisting in cells from pre-meiotic conditions? If so, what is known about the stability of Ndc80p?

3) Subsection “Two distinct *NDC80* transcript isoforms exist in meiosis”, last paragraph: provide statistical support that this decrease in the short form is significant. Do the same thing for the proposed increase in the level of the short form on deletion of its promoter (Figure 4).

4) Figure 4 and Figure 4—figure supplement 1: you need to provide an mRNA loading control.

5) Figure 5: Quantification of the mRNA from replicates with a loading control is needed to establish that the short *NUF2^ORF^*transcript is reduced in prophase relative to its level in pre-meiosis. How many repetitions are included in the quantification of the western data in panels B and D? Can the authors provide statistical information on reproducibility?

6) Figure 6: Quantification of the mRNA from replicates with a loading control is needed to establish that the level of *NDC80^ORF^* mRNA is not reduced during meiosis in the ime1 mutant, as expected from the absence of the long transcript.

7) Figure 6: In panel E, establish whether the reduction in *NDC80^ORF^* mRNA levels in the urs1 mutant is statistically significant. For panel H, explain why you don't see *NDC80^ORF^*mRNA levels increase in the urs1 mutant.

8) The promoter deletions remove large sections of the region upstream of the luti 5'-end. What information was used to guide the construction of these deletions? Is the region conserved between fungal species beyond the Ume6 and Ndt80 binding sites shown in Figure 6? While some of this information comes up in Figure 6, the authors should justify their choice of promoter deletions in Figure 4 by showing conservation of this region and highlighting putative transcription factor binding sites and core promoter elements. Also, where do these deletions map with respect to the 5' end of the luti mRNA? It would be valuable for the authors to include a supplemental figure that shows the nucleotide sequence for the 5' region of the *NDC80* gene. It would be helpful to highlight the nine AUG codons and show the predicted peptide sequence of the respective ORFs. Also, it would be helpful to highlight the MSE element and URS1 on this sequence and to denote the mutations used to alter the AUG codons and to introduce the stop codons. Relatedly, do any of the mutations impact the MSE? While the simplest model is that translation of the uORFs prevents downstream translation, if a mutation disrupted the MSE more complex models could be imagined.

9) Are the upstream AUG codons in the longer *NDC80^luti^* mRNA conserved through evolution? Is the presence of these precise AUG codons conserved in other fungal species? If not, do all fungi have additional AUG codons in the putative longer *NDC80* mRNAs? If these elements are not conserved in closely related species, the possibility of alternate mechanisms controlling the translation of the long *NDC80* mRNA could be considered.

---

## [Author Response]

Essential revisions:1) Figure 1: verify by western analysis that overexpression of Nuf2, Spc24, and Spc25 has actually occurred.

We have now verified the overexpression of Ndc80, Nuf2, Spc24, and Spc25 by western blot analysis. The data are shown in Figure 1—figure supplement 2.

2) Figure 2: analyze an unregulated mRNA as loading control, as the nuclease-resistant Pol I rRNA transcript is not the best control for establishing equal recovery of unstable Pol II mRNA transcripts.

As the reviewers suggest, it would be ideal to use a Pol II transcript as a loading control. However, this turns out to be a very difficult task for meiosis experiments, since the gene expression pattern is highly dynamic with more than ⅔ of genes undergoing more than 10-fold change in expression (Brar et al., 2012). Upon quantitative analysis of a published mRNA-seq dataset from a dense meiotic time course (Brar et al., 2012), we have been unable to find any Pol II transcripts that remain constant throughout meiosis. The analysis was performed as follows: For each gene, we calculated the deviation of the RPKM value for a given time point from the mean RPKM value across all meiotic time points. We then used a sliding window to determine whether the RPKM value for a given time point is within the desired range when compared to the mean (% change over mean RPKM). When the interval was set to 40%, there were only 22 genes that fulfill this criterion. All of these genes were expressed at low levels (max RPKM= 168, mean RPKM= 71). Among these genes, we tested the 6 with the highest mRNA abundance: *NPL6, MSB3, TSC10, COQ5, YMR31* and *GRX5*. We were only able to detect two of them by Northern, though the data quality was poor due to weak expression.

Next we tested four abundant genes (max RPKM = 7589, mean RPKM= 2617) that differ less than four fold across all time points: *PFY1, ZEO1, CIT1* and *SEC17*. Among these genes, *CIT1* was the most reasonable candidate as a Pol II loading control, in terms of its size and consistency. However, *CIT1* transcript still shows a noticeable variability, especially during the early time points (0-3h) and therefore cannot be used as a reliable loading control during these stages. It is also worth noting that the rRNA quality seems to be a reasonable estimate for the mRNA quality, since in the samples where we detect reduced or no signal with *rRNA*, the corresponding Pol ll transcript displays a very similar trend.

**Author response image 2. respfig2:** 

As an alternative loading control for the northern blots, we used a Pol III transcript, *SCR1*, the RNA subunit of the signal recognition particle. Although not a Pol II transcript, *SCR1* is less abundant than the rRNA and therefore can be detected in a more quantitative manner using radioactivity. In addition, it has been previously used as a loading control for meiosis (Frenk et al., 2014). Together, we think that this is the best possible loading control to use, in addition to rRNA, during a process in which very few, if any, Pol II transcribed mRNAs are expressed at a consistent level. In the experiments where thegels were run for a longer time, we used *CIT1* as a loading control instead of *SCR1* due to the larger molecular weight of *CIT1*.

Also for this figure, at early times in prophase, fairly high levels of Ndc80p are present, yet the primary transcript is clearly the luti mRNA. What is the source of this pool of Ndc80? Do the authors think it is persisting in cells from pre-meiotic conditions? If so, what is known about the stability of Ndc80p?

The reviewers bring up a very good point. We do think that the pool of Ndc80 protein in early meiosis is persisting from pre-meiotic conditions. Upon entry to meiotic prophase, Ndc80 is degraded through a regulated process. We have the following data that support this hypothesis:

a) Treatment with the proteasome inhibitor MG-132 causes high steady-state level of Ndc80 during meiotic prophase, suggesting that Ndc80 degradation is proteasome-dependent.

b) A small region at the N-terminal region of Ndc80 is required for its degradation. Deletion of this region results in high steady-state level of Ndc80 protein in meiotic prophase, but does not disrupt the luti-mRNA mediated repression.

c) Meiotic depletion of Aurora B kinase (Ipl1 in budding yeast) leads to high steady-state level of Ndc80 (Meyer et al., 2015), but does not alter the luti-mRNA mediated repression.

We are currently dissecting the mechanism of this post-translational regulation further for a follow-up manuscript under preparation.

3) Subsection “Two distinct NDC80 transcript isoforms exist in meiosis”, last paragraph: provide statistical support that this decrease in the short form is significant. Do the same thing for the proposed increase in the level of the short form on deletion of its promoter (Figure 4).

We have performed statistical analysis for all of our smFISH data. The detailed description of the statistical analyses can be found in Materials and methods. The p-values are displayed in all the figures that have smFISH data (Figure 2, Figure 2—figure supplement 6, Figure 4, and Figure 6).

4) Figure 4 and Figure 4—figure supplement 1: you need to provide an mRNA loading control.

We believe the reviewers are referring to Figure 4 and/or 4D (4C is smFISH data quantification). *CIT1* has been added to Figure 4 as a loading control (*SCR1* has ran off the gel in all the repeats of this experiment). For Figure 4, we hybridized these blots with a *CIT1* probe. However, these membranes are rather old and the data we collected for *CIT1* were not of publication quality. We think that the current northern blotting data in Figure 4 are still highly informative despite lacking a loading transcript. One can reliably compare the relative intensities of *NDC80^ORF^* to *NDC80^luti^* mRNA for a given lane. It is apparent that in both strains where one of the two *NDC80^luti^* loci is disrupted, this perturbation results in co-expression of both *NDC80^ORF^* and *NDC80^luti^* in meiotic prophase. Yet the luti-mRNA dependent repression of Ndc80 protein synthesis is only detected when the luti-mRNA is expressed from the same locus as the V5-epitope tagged protein coding ORF-mRNA isoform. *SCR1* loading control has been added to Figure 4—figure supplement 3 (previously listed as Figure 4—figure supplement 1).

5) Figure 5: Quantification of the mRNA from replicates with a loading control is needed to establish that the short NUF2^ORF^ transcript is reduced in prophase relative to its level in pre-meiosis. How many repetitions are included in the quantification of the western data in panels B and D? Can the authors provide statistical information on reproducibility?

As is, only a single repeat was included in the quantification. However, the experiment was performed twice, and the replicate northerns and westerns for both experiments are provided below. We prefer not to plot the mean of both repeats on the same graph as these time courses are dynamic, and although they are synchronous, the exact timing and meiotic kinetics can vary from repeat to repeat.

The data for the biological repeat of the experiment from Figure 5 are shown below.

**Author response image 3. respfig3:** 

The data for the biological repeat of the experiment from Figure 5 are shown below.

**Author response image 4. respfig4:** 

6) Figure 6: Quantification of the mRNA from replicates with a loading control is needed to establish that the level of NDC80^ORF^ mRNA is not reduced during meiosis in the ime1 mutant, as expected from the absence of the long transcript.

We have now included *SCR1* blots in Figure 6 for wild-type and the *ime1∆* strains. The quantification of *NDC80^ORF^* relative to the *SCR1* loading control are in Figure 6—figure supplement 1. Similar to above, only a single repeat was included in the quantification. Below is the quantification of a second repeat.

**Author response image 5. respfig5:** 

We removed the *ime4∆* data from this figure panel, since we obtained inconsistent data between biological repeats. We observed inconsistent kinetics between wild-type and *ime4∆* strains which precluded us from making reliable measurements of *NDC80^ORF^* transcript abundance.

7) Figure 6: In panel E, establish whether the reduction in NDC80^ORF^ mRNA levels in the urs1 mutant is statistically significant. For panel H, explain why you don't see NDC80^ORF^ mRNA levels increase in the urs1 mutant.

We have performed statistical analysis for Figure 6, and these data are now included in the corresponding figures.

Explanation for Figure 6: Based on our statistical analysis, the difference in the *NDC80^ORF^* transcript level between wild-type and *urs1∆* during meiotic prophase is not statistically significant, even though there was clear difference in northern blot analysis (Figure 6). We consider the possibility that our smFISH quantification method has a technical limitation when the *NDC80^luti^*isoform is highly expressed. Since we identified *NDC80^ORF^* based on the presence of the Q 670 signal (both transcripts) and the absence of CF 590 signal (*NDC80^luti^* unique probes), a missed localization in the CF 590 channel would cause us to over-estimate the number of *NDC80^ORF^*transcripts. In our control experiments using alternating probes (Figure 2—figure supplement 5), we measured that ~6% of the Q 670 spots lack colocalizing signal from the CF 590 channel. In conditions where the *NDC80^luti^* isoform is highly expressed, such as wild-type meiotic prophase, we expect to miss ~1 CF 590 spot per cell, which would then be interpreted as an extra *NDC80^ORF^*molecule. During meiotic prophase, the number of *NDC80^luti^* transcripts in wild-type is higher in comparison to that of the *urs1∆* mutant cells (the median of *NDC80^luti^* transcripts is 15 in wild-type, that of *urs1∆* is merely 5, Figure 6), therefore the number of mRNA molecules being mis-classified as *NDC80^ORF^* would also be higher in wild-type cells. Thus, it is possible that we have over-estimated the number of *NDC80^ORF^* transcripts in the wild-type strain. Given these limitations, we consider that the difference in the transcript level between the wildtype and *urs1∆* mutant is too subtle to be detected by our smFISH analysis.

8) The promoter deletions remove large sections of the region upstream of the luti 5'-end. What information was used to guide the construction of these deletions? Is the region conserved between fungal species beyond the Ume6 and Ndt80 binding sites shown in Figure 6? While some of this information comes up in Figure 6, the authors should justify their choice of promoter deletions in Figure 4 by showing conservation of this region and highlighting putative transcription factor binding sites and core promoter elements. Also, where do these deletions map with respect to the 5' end of the luti mRNA? It would be valuable for the authors to include a supplemental figure that shows the nucleotide sequence for the 5' region of the NDC80 gene. It would be helpful to highlight the nine AUG codons and show the predicted peptide sequence of the respective ORFs. Also, it would be helpful to highlight the MSE element and URS1 on this sequence and to denote the mutations used to alter the AUG codons and to introduce the stop codons. Relatedly, do any of the mutations impact the MSE? While the simplest model is that translation of the uORFs prevents downstream translation, if a mutation disrupted the MSE more complex models could be imagined.

To address the reviewers’ comments, we have generated two annotated nucleotide sequence files:

a) 5’ region of the *S. cerevisiae NDC80* gene along with the engineered mutations used in this study (Figure 4—figure supplement 1).

b) Clustal alignment of the 5’ intergenic regions from five fungal species (Figure 9—figure supplement 1 and Figure 9—figure supplement 2), as well as a table listing all the predicted peptide sequence for each uORF in each fungal species (Supplementary file 1). These files include all the information requested by the reviewers.

We identified the region required for *NDC80^luti^* mRNA expression empirically by systematically replacing different segments of the *NDC80* upstream intergenic region with a selectable marker: 1) ∆-600 to -300, 2) ∆-600 to -400, and 3) ∆-600 to -500. We chose these three deletions based on our conservation analysis of the Ume6 and Ndt80 binding sites. In these deletion constructs, we removed the predicted Ume6 binding site, but retain the Ndt80 binding site intact (See Figure 4—figure supplement 1). There are a few conserved regions based on the Clustal alignment, which may correspond to the core promoter and/or transcription factor binding sites. We currently don’t have further information about the specific identity of these regions. We corrected the text and figures and replaced “*NDC80^luti^* promoter deletion (∆p*NDC80^luti^*)” with “*NDC80^luti^* deletion (∆*NDC80^luti^*)” to more accurately describe the genotypic changes in the strains.

We don’t think that the *∆9AUG* and the *mini uORF* mutations affect the MSE site. They are also unlikely to regulate the Ndc80 protein levels through a mechanism that depends on the Ndt80 transcription factor. Our reasoning is as follows:

a) In both the *∆9AUG* mutant and the *mini uORF* mutants, both *NDC80^ORF^* mRNA and Ndc80 protein levels increase between 5.5 hr to 6.5 hr time points, which correspond to the timing of *NDT80* expression and activity (See Author response image 6 for a full time course of Figure 3). This suggests that the Ndt80-dependent wave of *NDC80^ORF^* mRNA and Ndc80 protein induction is unaltered in the *∆9AUG* and *mini uORF* mutant, consistent with a functional MSE sites in these mutants.

**Author response image 6. respfig6:** 

a) The *∆9AUG* mutant is unlikely to cause a high steady-state level of Ndc80 protein during meiotic prophase, in an *NDT80*-dependent manner. First, the expression and activity of Ndt80 protein is subject to the meiotic recombination checkpoint, which inhibits Ndt80 function until exit from meiotic prophase. Therefore, during the time when the Ndc80 protein levels are elevated in the *∆9AUG* mutant, Ndt80 is not expected to induce its target mRNAs, including *NDC80^ORF^*. Second, in the strains where the promoter of *NDT80* is replaced with *pGAL* promoter, Ndt80 is not expressed prior to *pGAL* induction by estradiol. In this case, even when Ndt80 is not expressed, high steady-state level of Ndc80 protein is still observed in the *∆9AUG* strain (See Author response image 7 for a time course comparing wild-type and *∆9AUG* in the absence of Ndt80 induction), suggesting that the Ndt80 transcription factor is not responsible for the elevated Ndc80 protein levels observed in the *∆9AUG* strain.

**Author response image 7. respfig7:** Wild-type (UB3380) versus *∆9AUG* (UB3683) cells were arrested in pachytene using the *pGAL-NDT80GAL4-ER* system. Cells were transferred to SPO at 0 hour. High level of Ndc80 protein was observed even in the absence of *GAL* induction.

b) Finally there is no visible expression of *NDC80^ORF^* mRNA by northern blots in the *∆9AUG* strain during meiotic prophase, despite high levels of Ndc80 protein. Combined with the observation that *NDC80^luti^* is the predominant transcript isoform in this stage, these data are most consistent with the interpretation that the long *NDC80* mRNA isoform is translated into Ndc80 protein in the absence of the AUG uORFs in its 5’ leader.

9) Are the upstream AUG codons in the longer *NDC80^luti^* mRNA conserved through evolution? Is the presence of these precise AUG codons conserved in other fungal species? If not, do all fungi have additional AUG codons in the putative longer NDC80 mRNAs? If these elements are not conserved in closely related species, the possibility of alternate mechanisms controlling the translation of the long NDC80 mRNA could be considered.

All five fungal species that we examined have at least 5 upstream AUG codons in the leader sequence of the putative *NDC80^luti^* mRNA (defined as the intergenic region at least 50 bps downstream of the predicted URS1 site up to the ORF translation start site). The exact number and locations of these AUG codons are not conserved, although three regions appear to be enriched for the presence of upstream AUGs (Figure 9—figure supplement 1 and Figure 9—figure supplement 2). Given that the *∆6AUG* mutant can readily repress the coding ability of *NDC80^luti^*, we consider that it only takes a few upstream AUGs to generate an un-decoded *NDC80^luti^*. The range of the upstream AUG codons (at least 5) among the five fungal species falls within this possibility. In addition, the sequences and the length of these uORFs also do not seem to be conserved (Supplementary file 1), consistent with the idea that the regulatory role of these uORFs come from the act of uORF translation, rather than the identity of the uORF peptides themselves. We have now discussed the results of this uORFs conservation analysis in our Discussion.